# Hardness of Learning Neural Networks with Natural Weights

**Amit Daniely**
School of Computer Science and Engineering, The Hebrew University, Jerusalem, Israel
and Google Research Tel-Aviv
`amit.daniely@mail.huji.ac.il`

**Gal Vardi**
Weizmann Institute of Science
`gal.vardi@weizmann.ac.il`

## Abstract

Neural networks are nowadays highly successful despite strong hardness results. The existing hardness results focus on the network architecture, and assume that the network's weights are arbitrary. A natural approach to settle the discrepancy is to assume that the network's weights are "well-behaved" and posses some generic properties that may allow efficient learning. This approach is supported by the intuition that the weights in real-world networks are not arbitrary, but exhibit some "random-like" properties with respect to some "natural" distributions. We prove negative results in this regard, and show that for depth-2 networks, and many "natural" weights distributions such as the normal and the uniform distribution, most networks are hard to learn. Namely, there is no efficient learning algorithm that is provably successful for most weights, and every input distribution. It implies that there is no generic property that holds with high probability in such random networks and allows efficient learning.

## 1 Introduction

Neural networks have revolutionized performance in multiple domains, such as computer vision and natural language processing, and have proven to be a highly effective tool for solving many challenging problems. This impressive practical success of neural networks is not well understood from the theoretical point of view. In particular, despite extensive research in recent years, it is not clear which models are learnable by neural networks algorithms.

Historically, there were many negative results for learning neural networks, and it is now known that under certain complexity assumptions, it is computationally hard to learn the class of functions computed by a neural network, even if the architecture is very simple. Indeed, it has been shown that learning neural networks is hard already for networks of depth 2 [35, 17]. These results hold already for *improper learning*, namely where the learning algorithm is allowed to return a hypothesis that does not belong to the considered hypothesis class.

In recent years, researchers have considered several ways to circumvent the discrepancy between those hardness results and the empirical success of neural networks. Namely, to understand which models are still learnable by neural networks algorithms. This effort includes proving learnability of linear models, including polynomials and kernel spaces [5, 47, 19, 15, 10, 32, 24, 39, 3, 4, 11, 48, 44, 29, 40, 8, 11, 34, 38, 36, 16], making assumptions on the input distribution [37, 9, 22, 23, 21, 31, 43], the network's weights [7, 43, 20, 2, 30], or both [33, 45].

In that respect, one fantastic result that can be potentially proven, is that neural networks are efficiently learnable if we assume that the network's weights are "well-behaved". Namely, that there are some generic properties of the network's weights that allow efficient learning. This approach is supported by the intuition that the weights in real-world networks are not arbitrary, but exhibit some "random-like" properties with respect to some "natural weights distributions" (e.g., where the weights are drawn from a normal distribution). We say that a property of the network's weights is a *natural property* with respect to such a natural weights distribution, if it holds with high probability. Existing hardness results focus on the network architecture, and assume that the weights are arbitrary. Thus, it is unclear whether there exists a natural property that allows efficient learning.

In this work, we investigate networks with random weights, and networks whose weights posses natural properties. We show that under various natural weights distributions *most networks are hard to learn*. Namely, there is no efficient learning algorithm that is provably successful for most weights, and every input distribution. We show that it implies that learning neural networks is hard already if their weights posses some natural property. Our hardness results are under the common assumption that refuting a random $K$-SAT formula is hard (the *RSAT assumption*). We emphasize that our results are valid for *any* learning algorithm, and not just common neural networks algorithms.

We consider networks of depth $2$ with a single output neuron, where the weights in the first layer are drawn from some natural distribution, and the weights in the second layer are all $1$. We consider multiple natural weights distributions, e.g., where the weights vector of each hidden neuron is distributed by a multivariate normal distribution, distributed uniformly on the sphere, or that each of its components is drawn i.i.d. from a normal, uniform or Bernoulli distribution. For each weights distribution, we show that learning such networks with high probability over the choice of the weights is hard. Thus, for such weights distributions, most networks are hard. It implies that there is no generic property that holds w.h.p. (e.g., with probability $0.9$) in such random networks and allows efficient learning. Hence, if generic properties that allow efficient learning exist, then they are not natural, namely, they are rare with respect to all the natural weights distributions that we consider.

We also consider random neural networks of depth $2$, where the first layer is a convolutional layer with non-overlapping patches such that its filter is drawn from some natural distribution, and the weights of the second layer are all $1$. We show that learning is hard also for such networks. It implies that there is no generic property that holds w.h.p. in such random convolutional networks and allows efficient learning.

### Related work

**Hardness of learning neural networks.** Hardness of learning neural networks in the standard (improper and distribution free) PAC model, follows from hardness of learning intersection of halfspaces. [35] showed that, assuming the hardness of the shortest vector problem, learning intersection of $n^\epsilon$ halfspaces for a constant $\epsilon > 0$ is hard. [17] showed that, under the RSAT assumption, learning intersection of $\omega(\log(n))$ halfspaces is hard. These results imply hardness of learning depth-2 networks with $n^\epsilon$ and $\omega(\log(n))$ hidden neurons (respectively). In the agnostic model, learning halfspaces is already hard [28, 14].

**Learning random neural networks.** [43] considers the problem of learning neural networks, where the weights are not arbitrary, but exhibit "nice" features such as non-degeneracy or some "random-like" appearance. The architecture of the networks that he considers is similar to ours. He shows that (under the RSAT assumption) no algorithm invariant to linear transformations can efficiently learn such networks if the columns of the weights matrix of the first layer are linearly independent. It implies that linear-invariant algorithms cannot learn such networks when the weights are chosen randomly. We note that this result holds only for linearly-invariant algorithms, which is a strong restriction. Standard gradient decent methods, for example, are not linearly invariant[1]. Our results hold for all algorithms.

In [20], it is shown that deep random neural networks (of depth $\omega(\log(n))$) with the sign activation function, are hard to learn in the statistical query (SQ) model. This result was recently extended by [2] to other activation functions, including the ReLU function. While their results hold for networks of depth $\omega(\log(n))$ and for SQ algorithms, our results hold for depth-2 networks and for all algorithms.

Our paper is structured as follows: In Section 2 we provide notations and definitions, followed by our results in Section 3. We sketch our proof ideas in Section 4, with all proofs deferred to Appendix A.

## 2 Preliminaries

### 2.1 Random Constraints Satisfaction Problems

Let $\mathcal{X}_{n,K}$ be the collection of *(signed) $K$-tuples*, that is, sequences $x = [(\alpha_1, i_1), \ldots, (\alpha_K, i_K)]$ for $\alpha_1, \ldots, \alpha_K \in \{\pm 1\}$ and distinct $i_1, \ldots, i_K \in [n]$. Each $x \in \mathcal{X}_{n,K}$ defines a function $U_x : \{\pm 1\}^n \to \{\pm 1\}^K$ by $U_x(\psi) = (\alpha_1 \psi_{i_1}, \ldots, \alpha_K \psi_{i_K})$.

Let $P : \{\pm 1\}^K \to \{0, 1\}$ be some predicate. A *$P$-constraint* with $n$ variables is a function $C : \{\pm 1\}^n \to \{0, 1\}$ of the form $C(x) = P \circ U_x$ for some $x \in \mathcal{X}_{n,K}$. An instance to the *CSP problem* $\mathrm{CSP}(P)$ is a *$P$-formula*, i.e., a collection $J = \{C_1, \ldots, C_m\}$ of $P$-constraints (each is specified by a $K$-tuple). The goal is to find an assignment $\psi \in \{\pm 1\}^n$ that maximizes the fraction of satisfied constraints (i.e., constraints with $C_i(\psi) = 1$). We will allow CSP problems where $P$ varies with $n$ (but is still fixed for every $n$). For example, we can look of the $\lceil \log(n) \rceil$-SAT problem.

We will consider the problem of distinguishing satisfiable from random $P$ formulas (a.k.a. the problem of refuting random $P$ formulas). For $m : \mathbb{N} \to \mathbb{N}$, we say that a randomized algorithm $\mathcal{A}$ efficiently solves the problem $\mathrm{CSP}_{m(n)}^{\mathrm{rand}}(P)$, if $\mathcal{A}$ is a polynomial-time algorithm such that: (1) If $J$ is a satisfiable instance to $\mathrm{CSP}(P)$ with $n$ variables and $m(n)$ constraints, then $\Pr(\mathcal{A}(J) = \text{"satisfiable"}) \geq \frac{3}{4} - o_n(1)$, where the probability is over the randomness of $\mathcal{A}$; (2) If $J$ is a random[2] instance to $\mathrm{CSP}(P)$ with $n$ variables and $m(n)$ constraints then $\Pr(\mathcal{A}(J) = \text{"random"}) \geq \frac{3}{4} - o_n(1)$, where the probability is over the choice of $J$ and the randomness of $\mathcal{A}$.

### 2.2 The random $K$-SAT assumption

Unless we face a dramatic breakthrough in complexity theory, it seems unlikely that hardness of learning can be established on standard complexity assumptions such as $\mathbf{P} \neq \mathbf{NP}$ (see [6, 18]). Indeed, all currently known lower bounds are based on assumptions from cryptography or average case hardness. Following [17] we will rely on an assumption about random $K$-SAT problems.

Let $J = \{C_1, \ldots, C_m\}$ be a random $K$-SAT formula on $n$ variables. Precisely, each $K$-SAT constraint $C_i$ is chosen independently and uniformly from the collection of $n$-variate $K$-SAT constraints. A simple probabilistic argument shows that for some constant $C$ (depending only on $K$), if $m \geq Cn$, then $J$ is not satisfiable w.h.p. The problem of *refuting random $K$-SAT formulas* (a.k.a. the problem of distinguishing satisfiable from random $K$-SAT formulas) is the problem $\mathrm{CSP}_{m(n)}^{\mathrm{rand}}(\mathrm{SAT}_K)$, where $\mathrm{SAT}_K$ is the predicate $z_1 \vee \ldots \vee z_K$.

The problem of refuting random $K$-SAT formulas has been extensively studied during the last 50 years. It is not hard to see that the problem gets easier as $m$ gets larger. The currently best known algorithms [27, 12, 13] can only refute random instances with $\Omega\left(n^{\lceil \frac{K}{2} \rceil}\right)$ constraints for $K \geq 4$ and $\Omega\left(n^{1.5}\right)$ constraints for $K = 3$. In light of that, [26] made the assumption that for $K = 3$, refuting random instances with $Cn$ constraints, for every constant $C$, is hard (and used that to prove hardness of approximation results). Here, we put forward the following assumption.

**Assumption 2.1.** *Refuting random $K$-SAT formulas with $n^{f(K)}$ constraints is hard for some $f(K) = \omega(1)$. Namely, for every $d > 0$ there is $K$ such that $\mathrm{CSP}_{n^d}^{\mathrm{rand}}(\mathrm{SAT}_K)$ is hard.*

**Terminology 2.2.** *A computational problem is* RSAT-hard *if its tractability refutes assumption 2.1.*

In addition to the performance of best known algorithms, there is plenty of evidence to the above assumption, in the form of hardness of approximation results, and lower bounds on various algorithms, including resolution, convex hierarchies, sum-of-squares, statistical algorithms, and more. We refer the reader to [17] for a more complete discussion.

## 2.3 Learning hypothesis classes and random neural networks

Let $\mathcal{H} \subseteq \mathbb{R}^{(\mathbb{R}^n)}$ be an hypothesis class. We say that a learning algorithm $\mathcal{L}$ *efficiently (PAC) learns* $\mathcal{H}$ if for every target function $f \in \mathcal{H}$ and every distribution $\mathcal{D}$ over $\mathbb{R}^n$, given only access to examples $(\mathbf{x}, f(\mathbf{x}))$ where $\mathbf{x} \sim \mathcal{D}$, the algorithm $\mathcal{L}$ runs in time polynomial in $n$ and returns with probability at least $\frac{9}{10}$ (over the internal randomness of $\mathcal{L}$), a predictor $h$ such that $\mathbb{E}_{\mathbf{x} \sim \mathcal{D}}\left[(h(\mathbf{x}) - f(\mathbf{x}))^2\right] \leq \frac{1}{10}$.

For a real matrix $W = (\mathbf{w}_1, \ldots, \mathbf{w}_m)$ of size $n \times m$, let $h_W : \mathbb{R}^n \to [0,1]$ be the function $h_W(\mathbf{x}) = \left[\sum_{i=1}^m [\langle \mathbf{w}_i, \mathbf{x} \rangle]_+\right]_{[0,1]}$, where $[z]_+ = \max\{0, z\}$ is the ReLU function, and $[z]_{[0,1]} = \min\{1, \max\{0, z\}\}$ is the clipping operation on the interval $[0,1]$. This corresponds to depth-2 networks with $m$ hidden neurons, with no bias in the first layer, and where the outputs of the first layer are simply summed and moved through a clipping non-linearity (this operation can also be easily implemented using a second layer composed of two ReLU neurons).

Let $\mathcal{D}_{\mathrm{mat}}$ be a distribution over real matrices of size $n \times m$. We assume that $m \leq n$. We say that a learning algorithm $\mathcal{L}$ *efficiently learns a random neural network* with respect to $\mathcal{D}_{\mathrm{mat}}$ ($\mathcal{D}_{\mathrm{mat}}$-random network, for short), if it satisfies the following property. For a random matrix $W$ drawn according to $\mathcal{D}_{\mathrm{mat}}$, and every distribution $\mathcal{D}$ (that may depend on $W$) over $\mathbb{R}^n$, given only access to examples $(\mathbf{x}, h_W(\mathbf{x}))$ where $\mathbf{x} \sim \mathcal{D}$, the algorithm $\mathcal{L}$ runs in time polynomial in $n$ and returns with probability at least $\frac{3}{4}$ over the choice of $W$ and the internal randomness of $\mathcal{L}$, a predictor $h$ such that

$$\mathbb{E}_{\mathbf{x} \sim \mathcal{D}}\left[(h(\mathbf{x}) - h_W(\mathbf{x}))^2\right] \leq \frac{1}{10}.$$

**Remark 2.1.** *Learning an hypothesis class requires that for every target function in the class and every input distribution the learning algorithm succeeds w.h.p., and learning a random neural network requires that for a random target function and every input distribution the learning algorithm succeeds w.h.p. Thus, in the former case an adversary chooses both the target function and the input distribution, and in the later the target function is chosen randomly and then the adversary chooses the input distribution. Therefore, the requirement from the algorithm for learning random neural networks is weaker then the requirement for learning neural networks in the standard PAC-learning model. We show hardness of learning already under this weaker requirement. Then, we show that hardness of learning random neural networks, implies hardness of learning neural networks with "natural" weights under the standard PAC-learning model.*

## 2.4 Notations and terminology

We denote by $\mathcal{U}([-r, r])$ the uniform distribution over the interval $[-r, r]$ in $\mathbb{R}$; by $\mathcal{U}(\{\pm r\})$ the symmetric Bernoulli distribution, namely, $Pr(r) = Pr(-r) = \frac{1}{2}$; by $\mathcal{N}(0, \sigma^2)$ the normal distribution with mean $0$ and variance $\sigma^2$, and by $\mathcal{N}(\mathbf{0}, \Sigma)$ the multivariate normal distribution with mean $\mathbf{0}$ and covariance matrix $\Sigma$. We say that a distribution over $\mathbb{R}$ is symmetric if it is continuous and its density satisfies $f(x) = f(-x)$ for every $x \in \mathbb{R}$, or that it is discrete and $Pr(x) = Pr(-x)$ for every $x \in \mathbb{R}$. For a matrix $M$ we denote by $s_{\min}(M)$ and $s_{\max}(M)$ the minimal and maximal singular values of $M$. For $\mathbf{x} \in \mathbb{R}^n$ we denote by $\|\mathbf{x}\|$ its $L_2$ norm. We denote the unit sphere in $\mathbb{R}^n$ by $\mathbb{S}^{n-1}$. For $t \in \mathbb{N}$ let $[t] = \{1, \ldots, t\}$. We say that an algorithm is efficient if it runs in polynomial time.

## 3 Results

We show RSAT-hardness for learning $\mathcal{D}_{\mathrm{mat}}$-random networks, where $\mathcal{D}_{\mathrm{mat}}$ corresponds either to a fully-connected layer, or to a convolutional layer. It implies hardness of learning depth-2 neural networks whose weights satisfy some natural property. We focus on the case of networks with $\mathcal{O}(\log^2(n))$ hidden neurons. We note, however, that our results can be extended to networks with $q(n)$ hidden neurons, for any $q(n) = \omega(\log(n))$. Moreover, while we consider networks whose weights in the second layer are all $1$, our results can be easily extended to networks where the weights in the second layer are arbitrary or random positive numbers.

### 3.1 Fully-connected neural networks

We start with random fully-connected neural networks. First, we consider a distribution $\mathcal{D}_{\mathrm{mat}}$ over real matrices, such that the entries are drawn i.i.d. from a symmetric distribution. We say that a random variable $z$ is *b-subgaussian* for some $b > 0$ if for all $t > 0$ we have $Pr\left(|z| > t\right) \leq 2\exp\left(-\frac{t^2}{b^2}\right)$.

**Theorem 3.1.** *Let $z$ be a symmetric random variable with variance $\sigma^2$. Assume that the random variable $z' = \frac{z}{\sigma}$ is b-subgaussian for some fixed b. Let $\epsilon > 0$ be a small constant, let $m = \mathcal{O}(\log^2(n))$, and let $\mathcal{D}_{\mathrm{mat}}$ be a distribution over $\mathbb{R}^{n \times m}$, such that the entries are i.i.d. copies of z. Then, learning a $\mathcal{D}_{\mathrm{mat}}$-random neural network is RSAT-hard, already if the distribution $\mathcal{D}$ is over vectors of norm at most $\frac{n^\epsilon}{\sigma}$ in $\mathbb{R}^n$.*

Since the normal distribution, the uniform distribution over an interval, and the symmetric Bernoulli distribution are subgaussian ([41]), we have the following corollary.

**Corollary 3.1.** *Let $\epsilon > 0$ be a small constant, let $m = \mathcal{O}(\log^2(n))$, and let $\mathcal{D}_{\mathrm{mat}}$ be a distribution over $\mathbb{R}^{n \times m}$, such that the entries are drawn i.i.d. from a distribution $\mathcal{D}_z$. If $\mathcal{D}_z = \mathcal{N}(0, \sigma^2)$ (respectively, $\mathcal{D}_z = \mathcal{U}([-r, r])$ or $\mathcal{D}_z = \mathcal{U}(\{\pm r\})$), then learning a $\mathcal{D}_{\mathrm{mat}}$-random neural network is RSAT-hard, already if the distribution $\mathcal{D}$ is over vectors of norm at most $\frac{n^\epsilon}{\sigma}$ (respectively, $\frac{n^\epsilon}{r}$) in $\mathbb{R}^n$.*

In the following theorem, we consider the case where $\mathcal{D}_{\mathrm{mat}}$ is such that each column is drawn i.i.d. from a multivariate normal distribution.

**Theorem 3.2.** *Let $\Sigma$ be a positive definite matrix of size $n \times n$, and let $\lambda_{\min}$ be its minimal eigenvalue. Let $\epsilon > 0$ be a small constant, let $m = \mathcal{O}(\log^2(n))$, and let $\mathcal{D}_{\mathrm{mat}}$ be a distribution over $\mathbb{R}^{n \times m}$, such that each column is drawn i.i.d. from $\mathcal{N}(\mathbf{0}, \Sigma)$. Then, learning a $\mathcal{D}_{\mathrm{mat}}$-random neural network is RSAT-hard, already if the distribution $\mathcal{D}$ is over vectors of norm at most $\frac{n^\epsilon}{\sqrt{\lambda_{\min}}}$ in $\mathbb{R}^n$.*

We also study the case where the distribution $\mathcal{D}_{\mathrm{mat}}$ is such that each column is drawn i.i.d. from the uniform distribution on the sphere of radius $r$ in $\mathbb{R}^n$.

**Theorem 3.3.** *Let $m = \mathcal{O}(\log^2(n))$ and let $\mathcal{D}_{\mathrm{mat}}$ be a distribution over $\mathbb{R}^{n \times m}$, such that each column is drawn i.i.d. from the uniform distribution over $r \cdot \mathbb{S}^{n-1}$. Then, learning a $\mathcal{D}_{\mathrm{mat}}$-random neural network is RSAT-hard, already if the distribution $\mathcal{D}$ is over vectors of norm at most $\mathcal{O}\left(\frac{n\sqrt{n}\log^4(n)}{r}\right)$ in $\mathbb{R}^n$.*

From the above theorems we have the following corollary, which shows that learning neural networks (in the standard PAC-learning model) is hard already if the weights satisfy some natural property.

**Corollary 3.2.** *Let $m = \mathcal{O}(\log^2(n))$, and let $\mathcal{D}_{\mathrm{mat}}$ be a distribution over $\mathbb{R}^{n \times m}$ from Theorems 3.2, 3.3, or from Corollary 3.1. Let $P$ be a property that holds with probability at least $\frac{9}{10}$ for a matrix $W$ drawn from $\mathcal{D}_{\mathrm{mat}}$. Let $\mathcal{H} = \{h_W : W \in \mathbb{R}^{n \times m}, W \text{ satisfies } P\}$ be an hypothesis class. Then, learning $\mathcal{H}$ is RSAT-hard, already if the distribution $\mathcal{D}$ is over vectors of norm bounded by the appropriate expression from Theorems 3.2, 3.3, or Corollary 3.1.*

The corollary follows easily from the following argument: Assume that $\mathcal{L}$ learns $\mathcal{H}$. If a matrix $W$ satisfies $P$ then for every distribution $\mathcal{D}$, given access to examples $(\mathbf{x}, h_W(\mathbf{x}))$ where $\mathbf{x} \sim \mathcal{D}$, the algorithm $\mathcal{L}$ returns w.p. at least $\frac{9}{10}$ a predictor $h$ such that $\mathbb{E}_{\mathbf{x} \sim \mathcal{D}}\left[(h(\mathbf{x}) - h_W(\mathbf{x}))^2\right] \leq \frac{1}{10}$. Now, let $W \sim \mathcal{D}_{\mathrm{mat}}$. Since $W$ satisfies $P$ with probability at least $\frac{9}{10}$, then for every distribution $\mathcal{D}$, given access to examples $(\mathbf{x}, h_W(\mathbf{x}))$ where $\mathbf{x} \sim \mathcal{D}$, the algorithm $\mathcal{L}$ returns w.p. at least $\frac{9}{10} \cdot \frac{9}{10} \geq \frac{3}{4}$ a predictor $h$ such that $\mathbb{E}_{\mathbf{x} \sim \mathcal{D}}\left[(h(\mathbf{x}) - h_W(\mathbf{x}))^2\right] \leq \frac{1}{10}$. Hence $\mathcal{L}$ learns a $\mathcal{D}_{\mathrm{mat}}$-random neural network. Note that this argument holds also if $\mathcal{D}$ is over vectors of a bounded norm.

### 3.2 Convolutional neural networks

We now turn to random Convolutional Neural Networks (CNN). Here, the distribution $\mathcal{D}_{\mathrm{mat}}$ corresponds to a random convolutional layer. Our convolutional layer has a very simple structure with non-overlapping patches. Let $t$ be an integer that divides $n$, and let $\mathbf{w} \in \mathbb{R}^t$. We denote by

$h_{\mathbf{w}}^n : \mathbb{R}^n \to [0,1]$ the CNN

$$h_{\mathbf{w}}^n(\mathbf{x}) = \left[ \sum_{i=1}^{\frac{n}{t}} [\langle \mathbf{w}, (x_{t(i-1)+1}, \ldots, x_{t \cdot i}) \rangle]_+ \right]_{[0,1]} .$$

Note that the $h_{\mathbf{w}}^n$ has $\frac{n}{t}$ hidden neurons. A random CNN corresponds to a random vector $\mathbf{w} \in \mathbb{R}^t$. Let $\mathcal{D}_{\text{vec}}$ be a distribution over $\mathbb{R}^t$. A $\mathcal{D}_{\text{vec}}$-random CNN with $m$ hidden neurons is the CNN $h_{\mathbf{w}}^{mt}$ where $\mathbf{w}$ is drawn from $\mathcal{D}_{\text{vec}}$. Note that every such a distribution over CNNs, can be expressed by an appropriate distribution $\mathcal{D}_{\text{mat}}$ over matrices. Our results hold also if we replace the second layer of $h_{\mathbf{w}}^n$ with a max-pooling layer (instead of summing and clipping).

We start with random CNNs, where each component in the weight vector is drawn i.i.d. from a symmetric distribution $\mathcal{D}_z$ over $\mathbb{R}$. In the following theorem, the function $f(t)$ bounds the concentration of $\mathcal{D}_z$, and is needed in order to bound the support of the distribution $\mathcal{D}$.

**Theorem 3.4.** *Let $n = (n'+1)\log^2(n')$. Let $\mathcal{D}_z^{n'+1}$ be a distribution over $\mathbb{R}^{n'+1}$ such that each component is drawn i.i.d. from a symmetric distribution $\mathcal{D}_z$ over $\mathbb{R}$. Let $f(t) > 0$ be a function such that $Pr_{z \sim \mathcal{D}_z}(|z| < f(t)) = o_t(\frac{1}{t})$. Then, learning a $\mathcal{D}_z^{n'+1}$-random CNN $h_{\mathbf{w}}^n$ with $\log^2(n') = \mathcal{O}(\log^2(n))$ hidden neurons is RSAT-hard, already if the distribution $\mathcal{D}$ is over vectors of norm at most $\frac{\log^2(n')}{f(n')}$ in $\mathbb{R}^n$.*

If $\mathcal{D}_z$ is the uniform distribution $\mathcal{U}([-r,r])$ then we can choose $f(t) = \frac{r}{t \log(t)}$, if it is the normal distribution $\mathcal{N}(0,\sigma^2)$ then we can choose $f(t) = \frac{\sigma}{t \log(t)}$, and if it is the symmetric Bernoulli distribution $\mathcal{U}(\{\pm r\})$ then we can choose $f(t) = r$. Thus, we obtain the following corollary.

**Corollary 3.3.** *Let $\mathcal{D}_{\text{vec}}$ be a distribution such that each component is drawn i.i.d. from a distribution $\mathcal{D}_z$ over $\mathbb{R}$. If $\mathcal{D}_z = \mathcal{N}(0,\sigma^2)$ (respectively, $\mathcal{D}_z = \mathcal{U}([-r,r])$, $\mathcal{D}_z = \mathcal{U}(\{\pm r\})$), then learning a $\mathcal{D}_{\text{vec}}$-random CNN $h_{\mathbf{w}}^n$ with $\mathcal{O}(\log^2(n))$ hidden neurons is RSAT-hard, already if $\mathcal{D}$ is over vectors of norm at most $\frac{n \log(n)}{\sigma}$ (respectively, $\frac{n \log(n)}{r}$, $\frac{\log^2(n)}{r}$) in $\mathbb{R}^n$.*

We also consider $h_{\mathbf{w}}^n$ where $\mathbf{w}$ is drawn from a multivariate normal distribution $\mathcal{N}(\mathbf{0}, \Sigma)$.

**Theorem 3.5.** *Let $\Sigma$ be a positive definite matrix of size $t \times t$, and let $\lambda_{\min}$ be its minimal eigenvalue. Let $n = \mathcal{O}(t \log^2(t))$. Then, learning a $\mathcal{N}(\mathbf{0}, \Sigma)$-random CNN $h_{\mathbf{w}}^n$ (with $\mathcal{O}(\log^2(n))$ hidden neurons) is RSAT-hard, already if the distribution $\mathcal{D}$ is over vectors of norm at most $\frac{n \log(n)}{\sqrt{\lambda_{\min}}}$ in $\mathbb{R}^n$.*

Finally, we consider $h_{\mathbf{w}}^n$ such that $\mathbf{w}$ is drawn from the uniform distribution over the sphere.

**Theorem 3.6.** *Let $\mathcal{D}_{\text{vec}}$ be the uniform distribution over the sphere of radius $r$ in $\mathbb{R}^t$. Let $n = \mathcal{O}(t \log^2(t))$. Then, learning a $\mathcal{D}_{\text{vec}}$-random CNN $h_{\mathbf{w}}^n$ (with $\mathcal{O}(\log^2(n))$ hidden neurons) is RSAT-hard, already if the distribution $\mathcal{D}$ is over vectors of norm at most $\frac{\sqrt{n} \log(n)}{r}$ in $\mathbb{R}^n$.*

Now, the following corollary follows easily (from the same argument as in Corollary 3.2), and shows that learning CNNs (in the standard PAC-learning model) is hard already if the weights satisfy some natural property.

**Corollary 3.4.** *Let $\mathcal{D}_{\text{vec}}$ be a distribution over $\mathbb{R}^t$ from Theorems 3.5, 3.6, or from Corollary 3.3. Let $n = \mathcal{O}(t \log^2(t))$. Let $P$ be a property that holds with probability at least $\frac{9}{10}$ for a vector $\mathbf{w}$ drawn from $\mathcal{D}_{\text{vec}}$. Let $\mathcal{H} = \{h_{\mathbf{w}}^n : \mathbf{w} \in \mathbb{R}^t, \ \mathbf{w} \text{ satisfies } P\}$ be an hypothesis class. Then, learning $\mathcal{H}$ is RSAT-hard, already if the distribution $\mathcal{D}$ is over vectors of norm bounded by the appropriate expression from Theorems 3.5, 3.6, or Corollary 3.3.*

**Remark 3.1** (Improving the bounds on the support of $\mathcal{D}$)**.** *In Appendix C, we prove by a simple scaling argument, that by increasing the number of hidden neurons in the CNN from $\mathcal{O}(\log^2(n))$ to $\mathcal{O}(n)$ we can improve the bounds on the support of $\mathcal{D}$. We show that for any constant $\epsilon > 0$, we can improve the bounds on $\mathcal{D}$ in Corollary 3.3 to $\frac{n^\epsilon}{\sigma}$ and $\frac{n^\epsilon}{r}$, the bound in Theorem 3.5 to $\frac{n^\epsilon}{\sqrt{\lambda_{\min}}}$, and the bound in Theorem 3.6 to $\frac{n^\epsilon}{r}$. It also implies that Corollary 3.4 holds for these improved bounds on $\mathcal{D}$ if $\mathcal{H}$ consists of CNNs with $\mathcal{O}(n)$ hidden neurons.*

## 4 Main proof ideas

Let $\mathcal{H} = \{h_W : W \in \mathbb{R}^{n \times k}\}$ where $k = \mathcal{O}(\log^2(n))$. By [17], learning $\mathcal{H}$ is hard. We want to reduce the problem of learning $\mathcal{H}$ to learning a $\mathcal{D}_{\text{mat}}$-random neural network for some fixed $\mathcal{D}_{\text{mat}}$. Let $W \in \mathbb{R}^{n \times k}$ and let $S = \{(\mathbf{x}_1, h_W(\mathbf{x}_1)), \ldots, (\mathbf{x}_m, h_W(\mathbf{x}_m))\}$ be a sample. Let $\mathcal{D}_{\text{mat}}^M$ be a distribution over the group of $n \times n$ invertible matrices, and let $M \sim \mathcal{D}_{\text{mat}}^M$. Consider the sample $S' = \{(\mathbf{x}_1', h_W(\mathbf{x}_1)), \ldots, (\mathbf{x}_m', h_W(\mathbf{x}_m))\}$ where for every $i \in [m]$ we have $\mathbf{x}_i' = (M^\top)^{-1}\mathbf{x}_i$. Since $W^\top \mathbf{x}_i = W^\top M^\top (M^\top)^{-1}\mathbf{x}_i = (MW)^\top \mathbf{x}_i'$, we have $h_W(\mathbf{x}_i) = h_{MW}(\mathbf{x}_i')$. Thus, $S' = \{(\mathbf{x}_1', h_{MW}(\mathbf{x}_1')), \ldots, (\mathbf{x}_m', h_{MW}(\mathbf{x}_m'))\}$. Note that $MW$ is a random matrix. Now, assume that there is an algorithm $\mathcal{L}'$ that learns successfully from $S'$. Consider the follow algorithm $\mathcal{L}$. Given a sample $S$, the algorithm $\mathcal{L}$ runs $\mathcal{L}'$ on $S'$, and returns the hypothesis $h(\mathbf{x}) = \mathcal{L}'(S')((M^\top)^{-1}\mathbf{x})$. It is not hard to show that $\mathcal{L}$ learns successfully from $S$. Since our goal is to reduce the problem of learning $\mathcal{H}$ to learning a $\mathcal{D}_{\text{mat}}$-random network where $\mathcal{D}_{\text{mat}}$ is a fixed distribution, we need $MW$ to be a $\mathcal{D}_{\text{mat}}$-random matrix. However, the distribution of $MW$ depends on both $\mathcal{D}_{\text{mat}}^M$ and $W$ (which is an unknown matrix).

Hence, the challenge is to find a reduction that translates a sample that is realizable by $h_W$ to a sample that is realizable by a $\mathcal{D}_{\text{mat}}$-random network, without knowing $W$. In order to obtain such a reduction, we proceed in two steps. First, we show that learning neural networks of the form $h_W$ where $W \in \mathbb{R}^{n \times k}$, is hard already if we restrict $W$ to a set of matrices with a special structure. Then, we show a distribution $\mathcal{D}_{\text{mat}}^M$ such that if $M \sim \mathcal{D}_{\text{mat}}^M$ and $W$ has the special structure, then $MW \sim \mathcal{D}_{\text{mat}}$. This property, as we showed, enables us to reduce the problem of learning such special-structure networks to the problem of learning $\mathcal{D}_{\text{mat}}$-random networks.

In order to obtain a special structure for $W$, consider the class $\mathcal{H}_{\text{sign}-\text{cnn}}^{n,k} = \{h_{\mathbf{w}}^n : \mathbf{w} \in \{\pm 1\}^{\frac{n}{k}}\}$. Note that the CNNs in $\mathcal{H}_{\text{sign}-\text{cnn}}^{n,k}$ have $k = \mathcal{O}(\log^2(n))$ hidden neurons. The networks in $\mathcal{H}_{\text{sign}-\text{cnn}}^{n,k}$ have three important properties: (1) They are CNNs; (2) Their patches are non-overlapping; (3) The components in the filter $\mathbf{w}$ are in $\{\pm 1\}$. Hardness of learning $\mathcal{H}_{\text{sign}-\text{cnn}}^{n,k}$ can be shown by a reduction from the RSAT problem. We defer the details of this reduction to the next sections. Let $W = (\mathbf{w}^1, \ldots, \mathbf{w}^k)$ be the matrix of size $n \times k$ that corresponds to $h_{\mathbf{w}}^n$, namely $h_W = h_{\mathbf{w}}^n$. Note that for every $i \in [k]$ we have $\left(\mathbf{w}_{\frac{n(i-1)}{k}+1}^i, \ldots, \mathbf{w}_{\frac{ni}{k}}^i\right) = \mathbf{w}$, and $\mathbf{w}_j^i = 0$ for every other $j \in [n]$.

We now show a distribution $\mathcal{D}_{\text{mat}}^M$ such that if $M \sim \mathcal{D}_{\text{mat}}^M$ and $W$ has a structure that corresponds to $\mathcal{H}_{\text{sign}-\text{cnn}}^{n,k}$, then $MW \sim \mathcal{D}_{\text{mat}}$. We start with the case where $\mathcal{D}_{\text{mat}}$ is a distribution over matrices such that each column is drawn i.i.d. from the uniform distribution on the sphere. We say that a matrix $M$ of size $n \times n$ is a *diagonal-blocks matrix* if

$$M = \begin{bmatrix} B^{11} & \cdots & B^{1k} \\ \vdots & \ddots & \vdots \\ B^{k1} & \cdots & B^{kk} \end{bmatrix}$$

where each block $B^{ij}$ is a diagonal matrix $\text{diag}(z_1^{ij}, \ldots, z_{\frac{n}{k}}^{ij})$. We denote $\mathbf{z}^{ij} = (z_1^{ij}, \ldots, z_{\frac{n}{k}}^{ij})$, and $\mathbf{z}^j = (\mathbf{z}^{1j}, \ldots, \mathbf{z}^{kj}) \in \mathbb{R}^n$. Note that for every $j \in [k]$, the vector $\mathbf{z}^j$ contains all the entries on the diagonals of blocks in the $j$-th column of blocks in $M$. Let $\mathcal{D}_{\text{mat}}^M$ be a distribution over diagonal-blocks matrices, such that the vectors $\mathbf{z}^j$ are drawn i.i.d. according to the uniform distribution on $\mathbb{S}^{n-1}$. Let $W$ be a matrix that corresponds to $h_{\mathbf{w}}^n \in \mathcal{H}_{\text{sign}-\text{cnn}}^{n,k}$. Note that the columns of $W' = MW$ are i.i.d. copies from the uniform distribution on $\mathbb{S}^{n-1}$. Indeed, denote $M^\top = (\mathbf{v}^1, \ldots, \mathbf{v}^n)$. Then, for every line index $i \in [n]$ we denote $i = (b-1)\left(\frac{n}{k}\right) + r$, where $b, r$ are integers and $1 \le r \le \frac{n}{k}$. Thus, $b$ is the line index of the block in $M$ that correspond to the $i$-th line in $M$, and $r$ is the line index within the block. Now, note that $W'_{ij} = \langle \mathbf{v}^i, \mathbf{w}^j \rangle = \langle \left(\mathbf{v}_{(j-1)\left(\frac{n}{k}\right)+1}^i, \ldots, \mathbf{v}_{j\left(\frac{n}{k}\right)}^i\right), \mathbf{w} \rangle = \langle (B_{r1}^{bj}, \ldots, B_{r\left(\frac{n}{k}\right)}^{bj}), \mathbf{w} \rangle = B_{rr}^{bj} \cdot \mathbf{w}_r = z_r^{bj} \cdot \mathbf{w}_r$. Since $\mathbf{w}_r \in \{\pm 1\}$, and since the uniform distribution on a sphere does not change by multiplying a subset of component by $-1$, then the $j$-th column of $W'$ has the same distribution as $\mathbf{z}^j$, namely, the uniform distribution over $\mathbb{S}^{n-1}$. Also, the columns of $W'$ are independent. Thus, $W' \sim \mathcal{D}_{\text{mat}}$.

The case where $\mathcal{D}_{\mathrm{mat}}$ is a distribution over matrices such that the entries are drawn i.i.d. from a symmetric distribution (such as $\mathcal{U}([-r,r])$, $\mathcal{N}(0,\sigma^2)$ or $\mathcal{U}(\{\pm r\})$) can be shown in a similar way. The result for the case where $\mathcal{D}_{\mathrm{mat}}$ is such that each columns is drawn i.i.d. from a multivariate normal distribution $\mathcal{N}(\mathbf{0},\Sigma)$ cannot be obtained in the same way, since a $\mathcal{N}(\mathbf{0},\Sigma)$ might be sensitive to multiplication of its component by $-1$. However, recall that a vector in $\mathbb{R}^n$ whose components are i.i.d. copies from $\mathcal{N}(0,1)$ has the distribution $\mathcal{N}(\mathbf{0},I_n)$. Now, since every multivariate normal distribution $\mathcal{N}(\mathbf{0},\Sigma)$ can be obtained from $\mathcal{N}(\mathbf{0},I_n)$ by a linear transformation, we are able to show hardness also for the case where $\mathcal{D}_{\mathrm{mat}}$ is such that each column is drawn i.i.d. from $\mathcal{N}(\mathbf{0},\Sigma)$.

Recall that in our reduction we translate $S = \{(\mathbf{x}_1, h_W(\mathbf{x}_1)), \ldots, (\mathbf{x}_m, h_W(\mathbf{x}_m))\}$ to $S' = \{(\mathbf{x}'_1, h_W(\mathbf{x}_1)), \ldots, (\mathbf{x}'_m, h_W(\mathbf{x}_m))\}$ where for every $i \in [m]$ we have $\mathbf{x}'_i = (M^\top)^{-1}\mathbf{x}_i$. Therefore, we need to show that our choice of $M$ is such that it is invertible with high probability. Also, since we want to show hardness already if the input distribution $\mathcal{D}$ is supported on a bounded domain, then we need to bound the norm of $\mathbf{x}'_i$, with high probability over the choice of $M$. This task requires a careful analysis of the spectral norm of $(M^\top)^{-1}$, namely, of $(s_{\min}(M))^{-1}$.

The proofs of the results on random CNNs follow the same ideas. The main difference is that in this case, instead of multiplying $\mathbf{x}_i$ by $(M^\top)^{-1}$, we multiply each patch in $\mathbf{x}_i$ by an appropriate matrix.

**Proof structure**

We start with a few definitions. We say that a sample $S = \{(\mathbf{x}_i, y_i)\}_{i=1}^m \in (\mathbb{R}^n \times \{0,1\})^m$ is *scattered* if $y_1, \ldots, y_m$ are independent fair coins (in particular, they are independent from $\mathbf{x}_1, \ldots, \mathbf{x}_m$). We say that $S$ is *contained in $A \subseteq \mathbb{R}^n$* if $\mathbf{x}_i \in A$ for every $i \in [m]$.

Let $\mathcal{A}$ be an algorithm whose input is a sample $S = \{(\mathbf{x}_i, y_i)\}_{i=1}^{m(n)} \in (\mathbb{R}^n \times \{0,1\})^{m(n)}$ and whose output is either "scattered" or "$\mathcal{H}$-realizable", where $\mathcal{H}$ is an hypothesis class. We say that $\mathcal{A}$ distinguishes size-$m$ $\mathcal{H}$-realizable samples from scattered samples if the following holds: (1) If the sample $S$ is scattered, then $\Pr\left(\mathcal{A}(S) = \text{"scattered"}\right) \geq \frac{3}{4} - o_n(1)$, where the probability is over the choice of $S$ and the randomness of $\mathcal{A}$; (2) If the sample $S$ satisfies $h(\mathbf{x}_i) = y_i$ for every $i \in [m]$ for some $h \in \mathcal{H}$, then $\Pr\left(\mathcal{A}(S) = \text{"$\mathcal{H}$-realizable"}\right) \geq \frac{3}{4} - o_n(1)$, where the probability is over the randomness of $\mathcal{A}$. We denote by $\mathrm{SCAT}_{m(n)}^A(\mathcal{H})$ the problem of distinguishing size-$m(n)$ $\mathcal{H}$-realizable samples that are contained in $A \subseteq \mathbb{R}^n$ from scattered samples.

Now, let $\mathcal{A}'$ be an algorithm whose input is a sample $S = \{(\mathbf{x}_i, y_i)\}_{i=1}^{m(n)} \in (\mathbb{R}^n \times \{0,1\})^{m(n)}$ and whose output is either "scattered" or "$\mathcal{D}_{\mathrm{mat}}$-realizable". We say that $\mathcal{A}'$ distinguishes size-$m$ $\mathcal{D}_{\mathrm{mat}}$-realizable samples from scattered samples if the following holds: (1) If the sample $S$ is scattered, then $\Pr\left(\mathcal{A}'(S) = \text{"scattered"}\right) \geq \frac{3}{4} - o_n(1)$, where the probability is over the choice of $S$ and the randomness of $\mathcal{A}'$; (2) If the sample $S$ satisfies $h_W(\mathbf{x}_i) = y_i$ for every $i \in [m]$, where $W$ is a random matrix drawn from $\mathcal{D}_{\mathrm{mat}}$, then $\Pr\left(\mathcal{A}'(S) = \text{"$\mathcal{D}_{\mathrm{mat}}$-realizable"}\right) \geq \frac{3}{4} - o_n(1)$, where the probability is over the choice of $W$ and the randomness of $\mathcal{A}'$. We denote by $\mathrm{SCAT}_{m(n)}^A(\mathcal{D}_{\mathrm{mat}})$ the problem of distinguishing size-$m(n)$ $\mathcal{D}_{\mathrm{mat}}$-realizable samples that are contained in $A \subseteq \mathbb{R}^n$ from scattered samples. In the case of random CNNs, we denote by $\mathrm{SCAT}_{m(n)}^A(\mathcal{D}_{\mathrm{vec}}, n)$ the problem of distinguishing size-$m(n)$ scattered samples that are contained in $A$, from samples that are realizable by a random CNN $h_{\mathbf{w}}^n$ where $\mathbf{w} \sim \mathcal{D}_{\mathrm{vec}}$.

Recall that $\mathcal{H}_{\mathrm{sign-cnn}}^{n,m} = \{h_{\mathbf{w}}^n : \mathbf{w} \in \{\pm 1\}^{\frac{n}{m}}\}$. As we described, hardness of learning $\mathcal{D}_{\mathrm{mat}}$-random neural networks where the distribution $\mathcal{D}$ is supported on a bounded domain, can be shown by first showing hardness of learning $\mathcal{H}_{\mathrm{sign-cnn}}^{n,m}$ with some $m = \mathcal{O}(\log^2(n))$, where the distribution $\mathcal{D}$ is supported on some $A' \subseteq \mathbb{R}^n$, and then reducing this problem to learning $\mathcal{D}_{\mathrm{mat}}$-random networks where the distribution $\mathcal{D}$ is supported on some $A \subseteq \mathbb{R}^n$. We can show RSAT-hardness of learning $\mathcal{H}_{\mathrm{sign-cnn}}^{n,m}$ by using the methodology of [17] as follows: First, show that if there is an efficient algorithm that learns $\mathcal{H}_{\mathrm{sign-cnn}}^{n,m}$ where the distribution $\mathcal{D}$ is supported on $A' \subseteq \mathbb{R}^n$, then there is a fixed $d$ and an efficient algorithm that solves $\mathrm{SCAT}_{n^d}^{A'}(\mathcal{H}_{\mathrm{sign-cnn}}^{n,m})$, and then show that for every fixed $d$, the problem $\mathrm{SCAT}_{n^d}^{A'}(\mathcal{H}_{\mathrm{sign-cnn}}^{n,m})$ is RSAT-hard.

Our proof follows a slightly different path than the one described above. First, we show that if there is an efficient algorithm that learns $\mathcal{D}_{\mathrm{mat}}$-random neural networks where the distribution $\mathcal{D}$ is

supported on $A \subseteq \mathbb{R}^n$, then there is a fixed $d$ and an efficient algorithm that solves $\text{SCAT}_{n^d}^A(\mathcal{D}_{\text{mat}})$. Then, we show that for every fixed $d$, the problem $\text{SCAT}_{n^d}^{A'}(\mathcal{H}_{\text{sign-cnn}}^{n,m})$ with some $A' \subseteq \mathbb{R}^n$, is RSAT-hard. Finally, for the required matrix distributions $\mathcal{D}_{\text{mat}}$ and sets $A$, we show a reduction from $\text{SCAT}_{n^d}^{A'}(\mathcal{H}_{\text{sign-cnn}}^{n,m})$ to $\text{SCAT}_{n^d}^A(\mathcal{D}_{\text{mat}})$. The main difference between this proof structure and the one described in the previous paragraph, is that here, for every distribution $\mathcal{D}_{\text{mat}}$ we need to show a reduction from $\text{SCAT}_{n^d}^{A'}(\mathcal{H}_{\text{sign-cnn}}^{n,m})$ to $\text{SCAT}_{n^d}^A(\mathcal{D}_{\text{mat}})$ (which are decision problems), and in the previous proof structure for every $\mathcal{D}_{\text{mat}}$ we need to show a reduction between learning $\mathcal{H}_{\text{sign-cnn}}^{n,m}$ and learning $\mathcal{D}_{\text{mat}}$-random networks. We chose this proof structure since here the proof for each $\mathcal{D}_{\text{mat}}$ is a reduction between decision problems, and thus we found the proofs to be simpler and cleaner this way. Other than this technical difference, both proof structures are essentially similar and follow the same ideas. The case of random CNNs is similar, except that here we show, for each distribution $\mathcal{D}_{\text{vec}}$ over vectors, a reduction from $\text{SCAT}_{n^d}^{A'}(\mathcal{H}_{\text{sign-cnn}}^{n,m})$ to $\text{SCAT}_{n^d}^A(\mathcal{D}_{\text{vec}}, n)$.

## Broader Impact

Not applicable as far as we can see (this is a purely theoretical paper).

## Funding disclosure

This research is partially supported by ISF grant 2258/19.

## Footnotes

[1][43] shows that gradient decent can become linearly invariant if it is preceded by a certain preconditioning step.

[2]To be precise, in a random formula with $n$ variable and $m$ constraints, the $K$-tuple defining each constraint is chosen uniformly, and independently from the other constraints.

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
