[Supplementary Material]

# A Proofs

## A.1 Learning $\mathcal{D}_{\mathrm{mat}}$-random networks is harder than $\mathrm{SCAT}_{n^d}^A(\mathcal{D}_{\mathrm{mat}})$

**Theorem A.1.** *Let $\mathcal{D}_{\mathrm{mat}}$ be a distribution over matrices. Assume that there is an algorithm that learns $\mathcal{D}_{\mathrm{mat}}$-random neural networks where the distribution $\mathcal{D}$ is supported on $A \subseteq \mathbb{R}^n$. Then, there is a fixed $d$ and an efficient algorithm that solves $\mathrm{SCAT}_{n^d}^A(\mathcal{D}_{\mathrm{mat}})$.*

*Proof.* Let $\mathcal{L}$ be an efficient learning algorithm that learns $\mathcal{D}_{\mathrm{mat}}$-random neural networks where the distribution $\mathcal{D}$ is supported on $A$. Let $m(n)$ be such that $\mathcal{L}$ uses a sample of size at most $m(n)$. Let $p(n) = 9m(n) + n$. Let $S = \{(\mathbf{x}_i, y_i)\}_{i=1}^{p(n)} \in (\mathbb{R}^n \times \{0,1\})^{p(n)}$ be a sample that is contained in $A$. We will show an efficient algorithm $\mathcal{A}$ that distinguishes whether $S$ is scattered or $\mathcal{D}_{\mathrm{mat}}$-realizable. This implies that the theorem holds for $d$ such that $n^d \geq p(n)$.

Given $S$, the algorithm $\mathcal{A}$ learns a function $h : \mathbb{R}^n \to \mathbb{R}$ by running $\mathcal{L}$ with an examples oracle that generates examples by choosing a random (uniformly distributed) example $(\mathbf{x}_i, y_i) \in S$. We denote $\ell_S(h) = \frac{1}{p(n)} \sum_{i \in [p(n)]} (h(\mathbf{x}_i) - y_i)^2$. Now, if $\ell_S(h) \leq \frac{1}{10}$, then $\mathcal{A}$ returns that $S$ is $\mathcal{D}_{\mathrm{mat}}$-realizable, and otherwise it returns that it is scattered. Clearly, the algorithm $\mathcal{A}$ runs in polynomial time. We now show that if $S$ is $\mathcal{D}_{\mathrm{mat}}$-realizable then $\mathcal{A}$ recognizes it with probability at least $\frac{3}{4}$, and that if $S$ is scattered then it also recognizes it with probability at least $\frac{3}{4}$.

Assume first that $S$ is $\mathcal{D}_{\mathrm{mat}}$-realizable. Let $\mathcal{D}_S$ be the uniform distribution over $\mathbf{x}_i \in \mathbb{R}^n$ from $S$. In this case, since $\mathcal{D}_S$ is supported on $A$, we are guaranteed that with probability at least $\frac{3}{4}$ over the choice of $W$ and the internal randomness of $\mathcal{L}$, we have $\ell_S(h) = \mathbb{E}_{\mathbf{x} \sim \mathcal{D}_S} \left[ (h(\mathbf{x}) - h_W(\mathbf{x}))^2 \right] \leq \frac{1}{10}$. Therefore, the algorithm returns "$\mathcal{D}_{\mathrm{mat}}$-realizable".

Now, assume that $S$ is scattered. Let $h : \mathbb{R}^n \to \mathbb{R}$ be the function returned by $\mathcal{L}$. Let $h' : \mathbb{R}^n \to \{0,1\}$ be the following function. For every $\mathbf{x} \in \mathbb{R}^n$, if $h(\mathbf{x}) \geq \frac{1}{2}$ then $h'(\mathbf{x}) = 1$, and otherwise $h'(\mathbf{x}) = 0$. Note that for every $(\mathbf{x}_i, y_i) \in S$, if $h'(\mathbf{x}_i) \neq y_i$ then $(h(\mathbf{x}_i) - y_i)^2 \geq \frac{1}{4}$. Therefore, $\ell_S(h) \geq \frac{1}{4} \ell_S(h')$. Let $C \subseteq [p(n)]$ be the set of indices of $S$ that were not observed by $\mathcal{L}$. Note that given $C$, the events $\{h'(\mathbf{x}_i) = y_i\}_{i \in C}$ are independent from one another, and each has probability $\frac{1}{2}$. By the Hoefding bound, we have that $h'(\mathbf{x}_i) \neq y_i$ for at most $\frac{1}{2} - \sqrt{\frac{\ln(n)}{n}}$ fraction of the indices in $C$ with probability at most

$$\exp\left( -\frac{2|C|\ln(n)}{n} \right) = \exp\left( -\frac{2(8m(n) + n)\ln(n)}{n} \right) \leq \exp\left( -2\ln(n) \right) = \frac{1}{n^2} \ .$$

Thus, $h'(\mathbf{x}_i) \neq y_i$ for at least $\frac{1}{2} - o_n(1)$ fraction of the indices in $C$ with probability at least $1 - o_n(1)$. Hence,

$$\ell_S(h) \geq \frac{1}{4} \ell_S(h') \geq \frac{1}{4} \cdot \frac{|C|}{p(n)} \left( \frac{1}{2} - o_n(1) \right) = \frac{1}{4} \cdot \frac{8m(n) + n}{9m(n) + n} \left( \frac{1}{2} - o_n(1) \right) \geq \frac{1}{9} - o_n(1) \ .$$

Therefore, for large enough $n$, with probability at least $\frac{3}{4}$ we have $\ell_S(h) > \frac{1}{10}$, and thus the algorithm returns "scattered". $\square$

## A.2 $\mathrm{SCAT}_{n^d}^A(\mathcal{H}_{\mathrm{sign-cnn}}^{n,m})$ is RSAT-hard

For a predicate $P : \{\pm 1\}^K \to \{0,1\}$ we denote by $\mathrm{CSP}(P, \neg P)$ the problem whose instances are collections, $J$, of constraints, each of which is either $P$ or $\neg P$ constraint, and the goal is to maximize the number of satisfied constraints. Denote by $\mathrm{CSP}_{m(n)}^{\mathrm{rand}}(P, \neg P)$ the problem of distinguishing[3] satisfiable from random formulas with $n$ variables and $m(n)$ constraints. Here, in a random formula, each constraint is chosen w.p. $\frac{1}{2}$ to be a uniform $P$ constraint and w.p. $\frac{1}{2}$ a uniform $\neg P$ constraint.

We will consider the predicate $T_{K,M} : \{0,1\}^{KM} \to \{0,1\}$ defined by

$$T_{K,M}(z) = (z_1 \vee \ldots \vee z_K) \wedge (z_{K+1} \vee \ldots \vee z_{2K}) \wedge \ldots \wedge \left( z_{(M-1)K+1} \vee \ldots \vee z_{MK} \right) \ .$$

We will need the following lemma from [17]. For an overview of its proof, see Appendix B.

**Lemma A.1.** *[17] Let $q(n) = \omega(\log(n))$ with $q(n) \leq \frac{n}{\log(n)}$, and let $d$ and $K$ be fixed integers. The problem $\mathrm{CSP}^{\mathrm{rand}}_{n^d}(\mathrm{SAT}_K)$ can be efficiently reduced to the problem $\mathrm{CSP}^{\mathrm{rand}}_{n^{d-1}}(T_{K,q(n)}, \neg T_{K,q(n)})$.*

In the following lemma, we use Lemma A.1 in order to show RSAT-hardness of $\mathrm{SCAT}^A_{n^d}(\mathcal{H}^{n,m}_{\mathrm{sign-cnn}})$ with some appropriate $m$ and $A$.

**Lemma A.2.** *Let $n = (n'+1)\log^2(n')$, and let $d$ be a fixed integer. The problem $\mathrm{SCAT}^A_{n^d}(\mathcal{H}^{n,\log^2(n')}_{\mathrm{sign-cnn}})$, where $A$ is the ball of radius $\log^2(n')$ in $\mathbb{R}^n$, is RSAT-hard.*

*Proof.* By Assumption 2.1, there is $K$ such that $\mathrm{CSP}^{\mathrm{rand}}_{(n')^{d+2}}(\mathrm{SAT}_K)$ is hard, where the $K$-SAT formula is over $n'$ variables. Then, by Lemma A.1, the problem $\mathrm{CSP}^{\mathrm{rand}}_{(n')^{d+1}}(T_{K,\log^2(n')}, \neg T_{K,\log^2(n')})$ is also hard. We will reduce $\mathrm{CSP}^{\mathrm{rand}}_{(n')^{d+1}}(T_{K,\log^2(n')}, \neg T_{K,\log^2(n')})$ to $\mathrm{SCAT}^A_{(n')^{d+1}}(\mathcal{H}^{n,\log^2(n')}_{\mathrm{sign-cnn}})$. Since $(n')^{d+1} > n^d$, it would imply that $\mathrm{SCAT}^A_{n^d}(\mathcal{H}^{n,\log^2(n')}_{\mathrm{sign-cnn}})$ is RSAT-hard.

Let $J = \{C_1, \ldots, C_{(n')^{d+1}}\}$ be an input for $\mathrm{CSP}^{\mathrm{rand}}_{(n')^{d+1}}(T_{K,\log^2(n')}, \neg T_{K,\log^2(n')})$. Namely, each constraint $C_i$ is either a CNF or a DNF formula. Equivalently, $J$ can be written as $J' = \{(C'_1, y_1), \ldots, (C'_{(n')^{d+1}}, y_{(n')^{d+1}})\}$ where for every $i$, if $C_i$ is a DNF formula then $C'_i = C_i$ and $y_i = 1$, and if $C_i$ is a CNF formula then $C'_i$ is the DNF obtained by negating $C_i$, and $y_i = 0$. Given $J'$ as above, we encode each DNF formula $C'_i$ (with $\log^2(n')$ clauses) as a vector $\mathbf{x}_i \in \mathbb{R}^n$ such that each clause $[(\alpha_1, i_1), \ldots, (\alpha_K, i_K)]$ in $C'_i$ (a signed $K$-tuple) is encoded by a vector $\mathbf{z} = (z_1, \ldots, z_{n'+1})$ as follows. First, we have $z_{n'+1} = -(K-1)$. Then, for every $1 \leq j \leq K$ we have $z_{i_j} = \alpha_j$, and for every variable $l$ that does not appear in the clause we have $z_l = 0$. Thus, for every $1 \leq l \leq n'$, the value of $z_l$ indicates whether the $l$-th variable appears in the clause as a positive literal, a negative literal, or does not appear. The encoding $\mathbf{x}_i$ of $C'_i$ is the concatenation of the encodings of its clauses.

Let $S = \{(\mathbf{x}_1, y_1), \ldots, (\mathbf{x}_{(n')^{d+1}}, y_{(n')^{d+1}})\}$. If $J$ is random then $S$ is scattered, since each constraint $C_i$ is with probability $\frac{1}{2}$ a DNF formula, and with probability $\frac{1}{2}$ a CNF formula, and this choice is independent of the choice of the literals in $C_i$. Assume now that $J$ is satisfiable by an assignment $\psi \in \{\pm 1\}^{n'}$. Let $\mathbf{w} = (\psi, 1) \in \{\pm 1\}^{n'+1}$. Note that $S$ is realizable by the CNN $h^n_{\mathbf{w}}$ with $\log^2(n')$ hidden neurons. Indeed, if $\mathbf{z} \in \mathbb{R}^{n'+1}$ is the encoding of a clause of $C'_i$, then $\langle \mathbf{z}, \mathbf{w} \rangle = 1$ if all the $K$ literals in the clause are satisfied by $\psi$, and otherwise $\langle \mathbf{z}, \mathbf{w} \rangle \leq -1$. Therefore, $h^n_{\mathbf{w}}(\mathbf{x}_i) = y_i$.

Note that by our construction, for every $i \in [(n')^{d+1}]$ we have for large enough $n'$

$$\|\mathbf{x}_i\| = \sqrt{\log^2(n')\,(K + (K-1)^2)} \leq \log(n') \cdot K \leq \log^2(n') \,.$$

$\square$

## A.3 Hardness of learning random fully-connected neural networks

Let $n = (n'+1)\log^2(n')$. We say that a matrix $M$ of size $n \times n$ is a *diagonal-blocks matrix* if

$$M = \begin{bmatrix} B^{11} & \cdots & B^{1\log^2(n')} \\ \vdots & \ddots & \vdots \\ B^{\log^2(n')1} & \cdots & B^{\log^2(n')\log^2(n')} \end{bmatrix}$$

where each block $B^{ij}$ is a diagonal matrix $\mathrm{diag}(z^{ij}_1, \ldots, z^{ij}_{n'+1})$. For every $1 \leq i \leq n'+1$ let $S_i = \{i + j(n'+1) : 0 \leq j \leq \log^2(n') - 1\}$. Let $M_{S_i}$ be the submatrix of $M$ obtained by selecting the rows and columns in $S_i$. Thus, $M_{S_i}$ is a matrix of size $\log^2(n') \times \log^2(n')$. For $\mathbf{x} \in \mathbb{R}^n$ let $\mathbf{x}_{S_i} \in \mathbb{R}^{\log^2(n')}$ be the restriction of $\mathbf{x}$ to the coordinates $S_i$.

**Lemma A.3.** *Let $M$ be a diagonal-blocks matrix. Then,*

$$s_{\min}(M) \geq \min_{1 \leq i \leq n'+1} s_{\min}(M_{S_i}) \,.$$

*Proof.* For every $\mathbf{x} \in \mathbb{R}^n$ with $\|\mathbf{x}\| = 1$ we have

$$
\begin{aligned}
\|M\mathbf{x}\|^2 &= \sum_{1 \leq i \leq n'+1} \|M_{S_i}\mathbf{x}_{S_i}\|^2 \geq \sum_{1 \leq i \leq n'+1} \left(s_{\min}(M_{S_i})\|\mathbf{x}_{S_i}\|\right)^2 \\
&\geq \min_{1 \leq i \leq n'+1} \left(s_{\min}(M_{S_i})\right)^2 \sum_{1 \leq i \leq n'+1} \|\mathbf{x}_{S_i}\|^2 = \left(\min_{1 \leq i \leq n'+1} \left(s_{\min}(M_{S_i})\right)^2\right)\|\mathbf{x}\|^2 \\
&= \min_{1 \leq i \leq n'+1} \left(s_{\min}(M_{S_i})\right)^2 .
\end{aligned}
$$

Hence, $s_{\min}(M) \geq \min_{1 \leq i \leq n'+1} s_{\min}(M_{S_i})$. $\qquad\square$

### A.3.1 Proof of Theorem 3.1

Let $M$ be a diagonal-blocks matrix, where each block $B^{ij}$ is a diagonal matrix $\mathrm{diag}(z_1^{ij}, \ldots, z_{n'+1}^{ij})$. Assume that for all $i, j, l$ the entries $z_l^{ij}$ are i.i.d. copies of a random variable $z$ that has a symmetric distribution $\mathcal{D}_z$ with variance $\sigma^2$. Also, assume that the random variable $z' = \frac{z}{\sigma}$ is $b$-subgaussian for some fixed $b$.

**Lemma A.4.**

$$
Pr\left(s_{\min}(M) \leq \frac{\sigma}{n'\log^2(n')}\right) = o_n(1) .
$$

*Proof.* Let $M' = \frac{1}{\sigma}M$. By Lemma A.3, we have

$$
s_{\min}(M') \geq \min_{1 \leq i \leq n'+1} s_{\min}(M'_{S_i}) . \tag{1}
$$

Note that for every $i$, all entries of the matrix $M'_{S_i}$ are i.i.d. copies of $z'$.

Now, we need the following theorem:

**Theorem A.2.** *[42] Let $\xi$ be a real random variable with expectation $0$ and variance $1$, and assume that $\xi$ is $b$-subgaussian for some $b > 0$. Let $A$ be an $n \times n$ matrix whose entries are i.i.d. copies of $\xi$. Then, for every $t \geq 0$ we have*

$$
Pr\left(s_{\min}(A) \leq \frac{t}{\sqrt{n}}\right) \leq Ct + c^n
$$

*where $C > 0$ and $c \in (0, 1)$ depend only on $b$.*

By Theorem A.2, since each matrix $M'_{S_i}$ is of size $\log^2(n') \times \log^2(n')$, we have for every $i \in [n'+1]$ that

$$
Pr\left(s_{\min}(M'_{S_i}) \leq \frac{t}{\log(n')}\right) \leq Ct + c^{\log^2(n')} .
$$

By choosing $t = \frac{1}{n'\log(n')}$ we have

$$
Pr\left(s_{\min}(M'_{S_i}) \leq \frac{1}{n'\log^2(n')}\right) \leq \frac{C}{n'\log(n')} + c^{\log^2(n')} .
$$

Then, by the union bound we have

$$
Pr\left(\min_{1 \leq i \leq n'+1} \left(s_{\min}(M'_{S_i})\right) \leq \frac{1}{n'\log^2(n')}\right) \leq \frac{C(n'+1)}{n'\log(n')} + c^{\log^2(n')}(n'+1) = o_n(1) .
$$

Combining this with $s_{\min}(M) = \sigma \cdot s_{\min}(M')$ and with Eq. 1, we have

$$
\begin{aligned}
Pr\left(s_{\min}(M) \leq \frac{\sigma}{n'\log^2(n')}\right) &= Pr\left(s_{\min}(M') \leq \frac{1}{n'\log^2(n')}\right) \\
&\leq Pr\left(\min_{1 \leq i \leq n'+1} \left(s_{\min}(M'_{S_i})\right) \leq \frac{1}{n'\log^2(n')}\right) = o_n(1) .
\end{aligned}
$$

$\qquad\square$

**Lemma A.5.** *Let $\mathcal{D}_{\mathrm{mat}}$ be a distribution over $\mathbb{R}^{n \times \log^2(n')}$ such that each entry is drawn i.i.d. from $\mathcal{D}_z$. Note that a $\mathcal{D}_{\mathrm{mat}}$-random network $h_W$ has $\log^2(n') = \mathcal{O}(\log^2(n))$ hidden neurons. Let $d$ be a fixed integer. Then, $\mathrm{SCAT}_{n^d}^A(\mathcal{D}_{\mathrm{mat}})$ is RSAT-hard, where $A$ is the ball of radius $\frac{n \log^2(n)}{\sigma}$ in $\mathbb{R}^n$.*

*Proof.* By Lemma A.2, the problem $\mathrm{SCAT}_{n^d}^{A'}(\mathcal{H}_{\mathrm{sign-cnn}}^{n,\log^2(n')})$ where $A'$ is the ball of radius $\log^2(n')$ in $\mathbb{R}^n$, is RSAT-hard. We will reduce this problem to $\mathrm{SCAT}_{n^d}^A(\mathcal{D}_{\mathrm{mat}})$. Given a sample $S = \{(\mathbf{x}_i, y_i)\}_{i=1}^{n^d} \in (\mathbb{R}^n \times \{0,1\})^{n^d}$ with $\|\mathbf{x}_i\| \leq \log^2(n')$ for every $i \in [n^d]$, we will, with probability $1 - o_n(1)$, construct a sample $S'$ that is contained in $A$, such that if $S$ is scattered then $S'$ is scattered, and if $S$ is $\mathcal{H}_{\mathrm{sign-cnn}}^{n,\log^2(n')}$-realizable then $S'$ is $\mathcal{D}_{\mathrm{mat}}$-realizable. Note that our reduction is allowed to fail with probability $o_n(1)$. Indeed, distinguishing scattered from realizable requires success with probability $\frac{3}{4} - o_n(1)$ and therefore reductions between such problems are not sensitive to a failure with probability $o_n(1)$.

Assuming that $M$ is invertible (note that by Lemma A.4 it holds with probability $1 - o_n(1)$), let $S' = \{(\mathbf{x}_1', y_1), \ldots, (\mathbf{x}_{n^d}', y_{n^d})\}$ where for every $i \in [n^d]$ we have $\mathbf{x}_i' = (M^\top)^{-1}\mathbf{x}_i$. Note that if $S$ is scattered then $S'$ is also scattered.

Assume that $S$ is realizable by the CNN $h_{\mathbf{w}}^n$ with $\mathbf{w} \in \{\pm 1\}^{n'+1}$. Let $W$ be the matrix of size $n \times \log^2(n')$ such that $h_W = h_{\mathbf{w}}^n$. Thus, $W = (\mathbf{w}^1, \ldots, \mathbf{w}^{\log^2(n')})$ where for every $i \in [\log^2(n')]$ we have $(\mathbf{w}_{(i-1)(n'+1)+1}^i, \ldots, \mathbf{w}_{i(n'+1)}^i) = \mathbf{w}$, and $\mathbf{w}_j^i = 0$ for every other $j \in [n]$. Let $W' = MW$. Note that $S'$ is realizable by $h_{W'}$. Indeed, for every $i \in [n^d]$ we have $y_i = h_{\mathbf{w}}^n(\mathbf{x}_i) = h_W(\mathbf{x}_i)$, and $W^\top \mathbf{x}_i = W^\top M^\top (M^\top)^{-1}\mathbf{x}_i = (W')^\top \mathbf{x}_i'$. Also, note that the entries of $W'$ are i.i.d. copies of $z$. Indeed, denote $M^\top = (\mathbf{v}^1, \ldots, \mathbf{v}^n)$. Then, for every line $i \in [n]$ we denote $i = (b-1)(n'+1) + r$, where $b, r$ are integers and $1 \leq r \leq n'+1$. Thus, $b$ is the line index of the block in $M$ that correspond to the $i$-th line in $M$, and $r$ is the line index within the block. Now, note that

$$
\begin{aligned}
W_{ij}' &= \langle \mathbf{v}^i, \mathbf{w}^j \rangle = \left\langle \left(\mathbf{v}_{(j-1)(n'+1)+1}^i, \ldots, \mathbf{v}_{j(n'+1)}^i\right), \mathbf{w} \right\rangle = \langle (B_{r1}^{bj}, \ldots, B_{r(n'+1)}^{bj}), \mathbf{w} \rangle \\
&= B_{rr}^{bj} \cdot \mathbf{w}_r = z_r^{bj} \cdot \mathbf{w}_r \ .
\end{aligned}
$$

Since $\mathcal{D}_z$ is symmetric and $\mathbf{w}_r \in \{\pm 1\}$, we have $W_{ij}' \sim \mathcal{D}_z$ independently from the other entries. Thus, $W' \sim \mathcal{D}_{\mathrm{mat}}$. Therefore, $h_{W'}$ is a $\mathcal{D}_{\mathrm{mat}}$-random network.

By Lemma A.4, we have with probability $1 - o_n(1)$ that for every $i \in [n^d]$,

$$
\begin{aligned}
\|\mathbf{x}_i'\| &= \left\|(M^\top)^{-1}\mathbf{x}_i\right\| \leq s_{\max}\left((M^\top)^{-1}\right)\|\mathbf{x}_i\| = \frac{1}{s_{\min}(M^\top)}\|\mathbf{x}_i\| = \frac{1}{s_{\min}(M)}\|\mathbf{x}_i\| \\
&\leq \frac{n' \log^2(n')}{\sigma} \log^2(n') \leq \frac{n \log^2(n)}{\sigma} \ .
\end{aligned}
$$

$\square$

Finally, Theorem 3.1 follows immediately from Theorem A.1 and the following lemma.

**Lemma A.6.** *Let $\mathcal{D}_{\mathrm{mat}}$ be a distribution over $\mathbb{R}^{\tilde{n} \times m}$ with $m = \mathcal{O}(\log^2(\tilde{n}))$, such that each entry is drawn i.i.d. from $\mathcal{D}_z$. Let $d$ be a fixed integer, and let $\epsilon > 0$ be a small constant. Then, $\mathrm{SCAT}_{\tilde{n}^d}^A(\mathcal{D}_{\mathrm{mat}})$ is RSAT-hard, where $A$ is the ball of radius $\frac{\tilde{n}^\epsilon}{\sigma}$ in $\mathbb{R}^{\tilde{n}}$.*

*Proof.* For integers $k, l$ we denote by $\mathcal{D}_{\mathrm{mat}}^{k,l}$ the distribution over $\mathbb{R}^{k \times l}$ such that each entry is drawn i.i.d. from $\mathcal{D}_z$. Let $c = \frac{2}{\epsilon}$, and let $\tilde{n} = n^c$. By Lemma A.5, the problem $\mathrm{SCAT}_{n^{cd}}^{A'}(\mathcal{D}_{\mathrm{mat}}^{n,m})$ is RSAT-hard, where $m = \mathcal{O}(\log^2(n))$, and $A'$ is the ball of radius $\frac{n \log^2(n)}{\sigma}$ in $\mathbb{R}^n$. We reduce this problem to $\mathrm{SCAT}_{\tilde{n}^d}^A(\mathcal{D}_{\mathrm{mat}}^{\tilde{n},m})$, where $A$ is the ball of radius $\frac{\tilde{n}^\epsilon}{\sigma}$ in $\mathbb{R}^{\tilde{n}}$. Note that $m = \mathcal{O}(\log^2(n)) = \mathcal{O}(\log^2(\tilde{n}))$.

Let $S = \{(\mathbf{x}_i, y_i)\}_{i=1}^{n^{cd}} \in (\mathbb{R}^n \times \{0,1\})^{n^{cd}}$ with $\|\mathbf{x}_i\| \leq \frac{n \log^2(n)}{\sigma}$. For every $i \in [n^{cd}]$, let $\mathbf{x}_i' \in \mathbb{R}^{\tilde{n}}$ be the vector obtained from $\mathbf{x}_i$ by padding it with zeros. Thus, $\mathbf{x}_i' = (\mathbf{x}_i, 0, \ldots, 0)$. Note that $n^{cd} = \tilde{n}^d$. Let $S' = \{(\mathbf{x}_i', y_i)\}_{i=1}^{\tilde{n}^d}$. If $S$ is scattered then $S'$ is also scattered. Note that if $S$ is realizable by $h_W$ then $S'$ is realizable by $h_{W'}$ where $W'$ is obtained from $W$ by appending $\tilde{n} - n$

arbitrary lines. Assume that $S$ is $\mathcal{D}_{\mathrm{mat}}^{n,m}$-realizable, that is, $W \sim \mathcal{D}_{\mathrm{mat}}^{n,m}$. Then, $S'$ is realizable by $h_{W'}$ where $W'$ is obtained from $W$ by appending lines such that each component is drawn i.i.d. from $\mathcal{D}_z$, and therefore, $S'$ is $\mathcal{D}_{\mathrm{mat}}^{\tilde{n},m}$-realizable. Finally, for every $i \in \tilde{n}^d$ we have

$$\|\mathbf{x}_i'\| = \|\mathbf{x}_i\| \leq \frac{n \log^2(n)}{\sigma} = \frac{\tilde{n}^{\frac{1}{c}} \log^2(\tilde{n}^{\frac{1}{c}})}{\sigma} \leq \frac{\tilde{n}^{\frac{2}{c}}}{\sigma} = \frac{\tilde{n}^\epsilon}{\sigma} \ .$$

$\square$

### A.3.2 Proof of Theorem 3.2

Let $\mathcal{D}_{\mathrm{mat}}$ be a distribution over $\mathbb{R}^{n \times m}$ with $m = \log^2(n)$, such that each entry is drawn i.i.d. from $\mathcal{N}(0,1)$. Let $d$ be a fixed integer. By Lemma A.6, we have that $\mathrm{SCAT}_{n^d}^A(\mathcal{D}_{\mathrm{mat}})$ is RSAT-hard, where $A$ is the ball of radius $n^\epsilon$ in $\mathbb{R}^n$. Let $(\mathcal{N}(0,1))^n$ be the distribution over $\mathbb{R}^n$ where each component is drawn i.i.d. from $\mathcal{N}(0,1)$. Recall that $(\mathcal{N}(0,1))^n = \mathcal{N}(\mathbf{0}, I_n)$ ([46]). Therefore, in the distribution $\mathcal{D}_{\mathrm{mat}}$, the columns are drawn i.i.d. from $\mathcal{N}(\mathbf{0}, I_n)$. Let $\mathcal{D}_{\mathrm{mat}}'$ be a distribution over $\mathbb{R}^{n \times m}$, such that each column is drawn i.i.d. from $\mathcal{N}(\mathbf{0}, \Sigma)$. By Theorem A.1, we need to show that $\mathrm{SCAT}_{n^d}^{A'}(\mathcal{D}_{\mathrm{mat}}')$ is RSAT-hard, where $A'$ is the ball of radius $\frac{n^\epsilon}{\sqrt{\lambda_{\min}}}$ in $\mathbb{R}^n$. We show a reduction from $\mathrm{SCAT}_{n^d}^A(\mathcal{D}_{\mathrm{mat}})$ to $\mathrm{SCAT}_{n^d}^{A'}(\mathcal{D}_{\mathrm{mat}}')$.

Let $S = \{(\mathbf{x}_i, y_i)\}_{i=1}^{n^d} \in (\mathbb{R}^n \times \{0,1\})^{n^d}$ be a sample. Let $\Sigma = U\Lambda U^\top$ be the spectral decomposition of $\Sigma$, and let $M = U\Lambda^{\frac{1}{2}}$. Recall that if $\mathbf{w} \sim \mathcal{N}(\mathbf{0}, I_n)$ then $M\mathbf{w} \sim \mathcal{N}(\mathbf{0}, \Sigma)$ ([46]). For every $i \in [n^d]$, let $\mathbf{x}_i' = (M^\top)^{-1}\mathbf{x}_i$, and let $S' = \{(\mathbf{x}_1', y_1), \ldots, (\mathbf{x}_{n^d}', y_{n^d})\}$. Note that if $S$ is scattered then $S'$ is also scattered. If $S$ is realizable by a $\mathcal{D}_{\mathrm{mat}}$-random network $h_W$, then let $W' = MW$. Note that $S'$ is realizable by $h_{W'}$. Indeed, for every $i \in [n^d]$ we have $(W')^\top \mathbf{x}_i' = W^\top M^\top (M^\top)^{-1} \mathbf{x}_i = W^\top \mathbf{x}_i$. Let $W = (\mathbf{w}_1, \ldots, \mathbf{w}_m)$ and let $W' = (\mathbf{w}_1', \ldots, \mathbf{w}_m')$. Since $W' = MW$ then $\mathbf{w}_j' = M\mathbf{w}_j$ for every $j \in [m]$. Now, since $W \sim \mathcal{D}_{\mathrm{mat}}$, we have for every $j$ that $\mathbf{w}_j \sim \mathcal{N}(\mathbf{0}, I_n)$ (i.i.d.). Therefore, $\mathbf{w}_j' = M\mathbf{w}_j \sim \mathcal{N}(\mathbf{0}, \Sigma)$, and thus $W' \sim \mathcal{D}_{\mathrm{mat}}'$. Hence, $S'$ is $\mathcal{D}_{\mathrm{mat}}'$-realizable.

We now bound the norms of the vectors $\mathbf{x}_i'$ in $S'$. Note that for every $i \in [n^d]$ we have

$$\|\mathbf{x}_i'\| = \left\|(M^\top)^{-1}\mathbf{x}_i\right\| = \left\|U\Lambda^{-\frac{1}{2}}\mathbf{x}_i\right\| = \left\|\Lambda^{-\frac{1}{2}}\mathbf{x}_i\right\| \leq \lambda_{\min}^{-\frac{1}{2}} \|\mathbf{x}_i\| \leq \lambda_{\min}^{-\frac{1}{2}} n^\epsilon \ .$$

### A.3.3 Proof of Theorem 3.3

Let $n = (n'+1)\log^2(n')$, and let $M$ be a diagonal-blocks matrix, where each block $B^{ij}$ is a diagonal matrix $\mathrm{diag}(z_1^{ij}, \ldots, z_{n'+1}^{ij})$. We denote $\mathbf{z}^{ij} = (z_1^{ij}, \ldots, z_{n'+1}^{ij})$, and $\mathbf{z}^j = (\mathbf{z}^{1j}, \ldots, \mathbf{z}^{\log^2(n')j}) \in \mathbb{R}^n$. Note that for every $j \in [\log^2(n')]$, the vector $\mathbf{z}^j$ contains all the entries on the diagonals of blocks in the $j$-th column of blocks in $M$. Assume that the vectors $\mathbf{z}^j$ are drawn i.i.d. according to the uniform distribution on $r \cdot \mathbb{S}^{n-1}$.

**Lemma A.7.** *For some universal constant $c' > 0$ we have*

$$Pr\left(s_{\min}(M) \leq \frac{c'r}{n'\sqrt{n'}\log^5(n')}\right) = o_n(1) \ .$$

*Proof.* Let $M' = \frac{\sqrt{n}}{r}M$. For every $j \in [\log^2(n')]$, let $\tilde{\mathbf{z}}^j \in \mathbb{R}^n$ be the vector that contains all the entries on the diagonals of blocks in the $j$-th column of blocks in $M'$. That is, $\tilde{\mathbf{z}}^j = \frac{\sqrt{n}}{r}\mathbf{z}^j$. Note that the vectors $\tilde{\mathbf{z}}^j$ are i.i.d. copies from the uniform distribution on $\sqrt{n} \cdot \mathbb{S}^{n-1}$. By Lemma A.3, we have

$$s_{\min}(M') \geq \min_{1 \leq i \leq n'+1} s_{\min}(M_{S_i}') \ . \tag{2}$$

Note that for every $i$, all columns of the matrix $M_{S_i}'$ are projections of the vectors $\tilde{\mathbf{z}}^j$ on the $S_i$ coordinated. That is, the $j$-th column in $M_{S_i}'$ is obtained by drawing $\tilde{\mathbf{z}}^j$ from the uniform distribution on $\sqrt{n} \cdot \mathbb{S}^{n-1}$ and projecting on the coordinates $S_i$.

We say that a distribution is *isotropic* if it has mean zero and its covariance matrix is the identity. The covariance matrix of the uniform distribution on $\mathbb{S}^{n-1}$ is $\frac{1}{n}I_n$. Therefore, the uniform distribution on $\sqrt{n} \cdot \mathbb{S}^{n-1}$ is isotropic. We will need the following theorem.

**Theorem A.3.** *[1] Let $m \geq 1$ and let $A$ be an $m \times m$ matrix with independent columns drawn from an isotropic log-concave distribution. For every $\epsilon \in (0, 1)$ we have*

$$Pr\left(s_{\min}(A) \leq \frac{c\epsilon}{\sqrt{m}}\right) \leq Cm\epsilon$$

*where $c$ and $C$ are positive universal constants.*

We show that the distribution of the columns of $M'_{S_i}$ is isotropic and log-concave. First, since the uniform distribution on $\sqrt{n} \cdot \mathbb{S}^{n-1}$ is isotropic, then its projection on a subset of coordinates is also isotropic, and thus the distribution of the columns of $M'_{S_i}$ is isotropic. In order to show that it is log-concave, we analyze its density. Let $\mathbf{x} \in \mathbb{R}^n$ be a random variable whose distribution is the projection of a uniform distribution on $\mathbb{S}^{n-1}$ on $k$ coordinates. It is known that the probability density of $\mathbf{x}$ is (see [25])

$$f_{\mathbf{x}}(x_1, \ldots, x_k) = \frac{\Gamma(n/2)}{\Gamma((n-k)/2)\pi^{k/2}} \left(1 - \sum_{1 \leq i \leq k} x_i^2\right)^{\frac{n-k}{2} - 1},$$

where $\sum_{1 \leq i \leq k} x_i^2 < 1$. Recall that the columns of $M'_{S_i}$ are projections of the uniform distribution over $\sqrt{n} \cdot \mathbb{S}^{n-1}$, namely, the sphere of radius $\sqrt{n}$ and not the unit sphere. Thus, let $\mathbf{x}' = \sqrt{n}\mathbf{x}$. The probability density of $\mathbf{x}'$ is

$$
\begin{aligned}
f_{\mathbf{x}'}(x_1', \ldots, x_k') &= \frac{1}{(\sqrt{n})^k} f_{\mathbf{x}}\left(\frac{x_1'}{\sqrt{n}}, \ldots, \frac{x_k'}{\sqrt{n}}\right) \\
&= \frac{1}{n^{k/2}} \cdot \frac{\Gamma(n/2)}{\Gamma((n-k)/2)\pi^{k/2}} \left(1 - \sum_{1 \leq i \leq k} \left(\frac{x_i'}{\sqrt{n}}\right)^2\right)^{\frac{n-k}{2} - 1},
\end{aligned}
$$

where $\sum_{1 \leq i \leq k}(x_i')^2 < n$. We denote

$$g(n, k) = \frac{1}{n^{k/2}} \cdot \frac{\Gamma(n/2)}{\Gamma((n-k)/2)\pi^{k/2}}.$$

By replacing $k$ with $\log^2(n')$ we have

$$f_{\mathbf{x}'}(x_1', \ldots, x_{\log^2(n')}') = g(n, \log^2(n')) \left(1 - \frac{1}{n} \sum_{1 \leq i \leq \log^2(n')}(x_i')^2\right)^{\frac{n - \log^2(n')}{2} - 1}.$$

Hence, we have

$$
\begin{aligned}
\log f_{\mathbf{x}'}(x_1', \ldots, x_{\log^2(n')}') &= \\
\log\left(g(n, \log^2(n'))\right) &+ \left(\frac{n - \log^2(n')}{2} - 1\right) \cdot \log\left(1 - \frac{1}{n} \sum_{1 \leq i \leq \log^2(n')}(x_i')^2\right).
\end{aligned}
$$

Since $\frac{n - \log^2(n')}{2} - 1 > 0$, we need to show that the function

$$\log\left(1 - \frac{1}{n} \sum_{1 \leq i \leq \log^2(n')}(x_i')^2\right) \tag{3}$$

(where $\sum_{1 \leq i \leq \log^2(n')}(x_i')^2 < n$) is concave. This function can be written as $h(f(x_1, \ldots, x_{\log^2(n')}))$, where

$$h(x) = \log(1 + x),$$

$$f(x_1', \ldots, x_{\log^2(n')}') = -\frac{1}{n} \sum_{1 \leq i \leq \log^2(n')}(x_i')^2.$$

Recall that if $h$ is concave and non-decreasing, and $f$ is concave, then their composition is also concave. Clearly, $h$ and $f$ satisfy these conditions, and thus the function in Eq. 3 is concave. Hence $f_{\mathbf{x}'}$ is log-concave.

We now apply Theorem A.3 on $M'_{S_i}$, and obtain that for every $\epsilon \in (0,1)$ we have

$$Pr\left(s_{\min}(M'_{S_i}) \leq \frac{c\epsilon}{\log(n')}\right) \leq C\log^2(n')\epsilon .$$

By choosing $\epsilon = \frac{1}{n'\log^3(n')}$ we have

$$Pr\left(s_{\min}(M'_{S_i}) \leq \frac{c}{n'\log^4(n')}\right) \leq \frac{C}{n'\log(n')} .$$

Now, by the union bound

$$Pr\left(\min_{1\leq i\leq n'+1}(s_{\min}(M'_{S_i})) \leq \frac{c}{n'\log^4(n')}\right) \leq \frac{C}{n'\log(n')} \cdot (n'+1) = o_n(1) .$$

Combining this with $s_{\min}(M) = \frac{r}{\sqrt{n}}s_{\min}(M')$ and with Eq. 2, we have

$$\begin{aligned}
Pr\left(s_{\min}(M) \leq \frac{cr}{\sqrt{n}\cdot n'\log^4(n')}\right) &= Pr\left(s_{\min}(M') \leq \frac{c}{n'\log^4(n')}\right) \\
&\leq Pr\left(\min_{1\leq i\leq n'+1}(s_{\min}(M'_{S_i})) \leq \frac{c}{n'\log^4(n')}\right) = o_n(1) .
\end{aligned}$$

Note that

$$\frac{cr}{\sqrt{n}\cdot n'\log^4(n')} = \frac{cr}{\sqrt{n'+1}\cdot n'\log^5(n')} \geq \frac{cr}{2\sqrt{n'}\cdot n'\log^5(n')} = \frac{c'r}{\sqrt{n'}\cdot n'\log^5(n')} ,$$

where $c' = \frac{c}{2}$. Thus,

$$Pr\left(s_{\min}(M) \leq \frac{c'r}{\sqrt{n'}\cdot n'\log^5(n')}\right) \leq Pr\left(s_{\min}(M) \leq \frac{cr}{\sqrt{n}\cdot n'\log^4(n')}\right) = o_n(1) .$$

$\square$

Let $\mathcal{D}_{\mathrm{mat}}$ be a distribution over $\mathbb{R}^{n\times\log^2(n')}$ such that each column is drawn i.i.d. from the uniform distribution on $r\cdot\mathbb{S}^{n-1}$. Note that a $\mathcal{D}_{\mathrm{mat}}$-random network $h_W$ has $\log^2(n') = \mathcal{O}(\log^2(n))$ hidden neurons. Now, Theorem 3.3 follows immediately from Theorem A.1 and the following lemma.

**Lemma A.8.** *Let $d$ be a fixed integer. Then, $\mathrm{SCAT}_{n^d}^A(\mathcal{D}_{\mathrm{mat}})$ is RSAT-hard, where $A$ is a ball of radius $\mathcal{O}\left(\frac{n\sqrt{n}\log^4(n)}{r}\right)$ in $\mathbb{R}^n$.*

*Proof.* By Lemma A.2, the problem $\mathrm{SCAT}_{n^d}^{A'}(\mathcal{H}_{\mathrm{sign-cnn}}^{n,\log^2(n')})$ where $A'$ is the ball of radius $\log^2(n')$ in $\mathbb{R}^n$, is RSAT-hard. We will reduce this problem to $\mathrm{SCAT}_{n^d}^A(\mathcal{D}_{\mathrm{mat}})$. Given a sample $S = \{(\mathbf{x}_i, y_i)\}_{i=1}^{n^d} \in (\mathbb{R}^n \times \{0,1\})^{n^d}$ with $\|\mathbf{x}_i\| \leq \log^2(n')$ for every $i \in [n^d]$, we will, with probability $1 - o_n(1)$, construct a sample $S'$ that is contained in $A$, such that if $S$ is scattered then $S'$ is scattered, and if $S$ is $\mathcal{H}_{\mathrm{sign-cnn}}^{n,\log^2(n')}$-realizable then $S'$ is $\mathcal{D}_{\mathrm{mat}}$-realizable. Note that our reduction is allowed to fail with probability $o_n(1)$. Indeed, distinguishing scattered from realizable requires success with probability $\frac{3}{4} - o_n(1)$ and therefore reductions between such problems are not sensitive to a failure with probability $o_n(1)$.

Assuming that $M$ is invertible (by Lemma A.7 it holds with probability $1 - o_n(1)$), let $S' = \{(\mathbf{x}'_1, y_1), \ldots, (\mathbf{x}'_{n^d}, y_{n^d})\}$ where for every $i$ we have $\mathbf{x}'_i = (M^\top)^{-1}\mathbf{x}_i$. Note that if $S$ is scattered then $S'$ is also scattered.

Assume that $S$ is realizable by the CNN $h_{\mathbf{w}}^n$ with $\mathbf{w} \in \{\pm 1\}^{n'+1}$. Let $W$ be the matrix of size $n \times \log^2(n')$ such that $h_W = h_{\mathbf{w}}^n$. Thus, $W = (\mathbf{w}^1, \ldots, \mathbf{w}^{\log^2(n')})$ where for every $i \in [\log^2(n')]$

we have $(\mathbf{w}^i_{(i-1)(n'+1)+1}, \ldots, \mathbf{w}^i_{i(n'+1)}) = \mathbf{w}$, and $\mathbf{w}^i_j = 0$ for every other $j \in [n]$. Let $W' = MW$. Note that $S'$ is realizable by $h_{W'}$. Indeed, for every $i \in [n^d]$ we have $y_i = h^n_{\mathbf{w}}(\mathbf{x}_i) = h_W(\mathbf{x}_i)$, and $W^\top \mathbf{x}_i = W^\top M^\top (M^\top)^{-1} \mathbf{x}_i = (W')^\top \mathbf{x}'_i$. Also, note that the columns of $W'$ are i.i.d. copies from the uniform distribution on $r \cdot \mathbb{S}^{n-1}$. Indeed, denote $M^\top = (\mathbf{v}^1, \ldots, \mathbf{v}^n)$. Then, for every line index $i \in [n]$ we denote $i = (b-1)(n'+1) + r$, where $b, r$ are integers and $1 \leq r \leq n'+1$. Thus, $b$ is the line index of the block in $M$ that correspond to the $i$-th line in $M$, and $r$ is the line index within the block. Now, note that

$$
\begin{aligned}
W'_{ij} &= \langle \mathbf{v}^i, \mathbf{w}^j \rangle = \left\langle \left( \mathbf{v}^i_{(j-1)(n'+1)+1}, \ldots, \mathbf{v}^i_{j(n'+1)} \right), \mathbf{w} \right\rangle = \langle (B^{bj}_{r1}, \ldots, B^{bj}_{r(n'+1)}), \mathbf{w} \rangle \\
&= B^{bj}_{rr} \cdot \mathbf{w}_r = z^{bj}_r \cdot \mathbf{w}_r .
\end{aligned}
$$

Since $\mathbf{w}_r \in \{\pm 1\}$, and since the uniform distribution on a sphere does not change by multiplying a subset of component by $-1$, then the $j$-th column of $W'$ has the same distribution as $\mathbf{z}^j$, namely, the uniform distribution over $r \cdot \mathbb{S}^{n-1}$. Also, the columns of $W'$ are independent. Thus, $W' \sim \mathcal{D}_{\mathrm{mat}}$, and therefore $h_{W'}$ is a $\mathcal{D}_{\mathrm{mat}}$-random network.

By Lemma A.7, we have with probability $1 - o_n(1)$ that for every $i$,

$$
\begin{aligned}
\|\mathbf{x}'_i\| &= \|(M^\top)^{-1} \mathbf{x}_i\| \leq s_{\max}\left((M^\top)^{-1}\right) \|\mathbf{x}_i\| = \frac{1}{s_{\min}(M^\top)} \|\mathbf{x}_i\| = \frac{1}{s_{\min}(M)} \|\mathbf{x}_i\| \\
&\leq \frac{n'\sqrt{n'}\log^5(n')}{c'r} \cdot \log^2(n') \leq \frac{n\sqrt{n}\log^4(n)}{c'r} .
\end{aligned}
$$

Thus, $\|\mathbf{x}'_i\| = \mathcal{O}\left( \frac{n\sqrt{n}\log^4(n)}{r} \right)$. $\qquad\square$

## A.4   Hardness of learning random convolutional neural networks

### A.4.1   Proof of Theorem 3.4

Theorem 3.4 follows immediately from Theorem A.1 and the following lemma:

**Lemma A.9.** *Let $d$ be a fixed integer. Then,* $\mathrm{SCAT}^A_{n^d}(\mathcal{D}^{n'+1}_z, n)$ *is RSAT-hard, where $A$ is the ball of radius $\frac{\log^2(n')}{f(n')}$ in $\mathbb{R}^n$.*

*Proof.* By Lemma A.2, the problem $\mathrm{SCAT}^{A'}_{n^d}(\mathcal{H}^{n,\log^2(n')}_{\mathrm{sign-cnn}})$ where $A'$ is the ball of radius $\log^2(n')$ in $\mathbb{R}^n$, is RSAT-hard. We will reduce this problem to $\mathrm{SCAT}^A_{n^d}(\mathcal{D}^{n'+1}_z, n)$. Given a sample $S = \{(\mathbf{x}_i, y_i)\}^{n^d}_{i=1} \in (\mathbb{R}^n \times \{0,1\})^{n^d}$ with $\|\mathbf{x}_i\| \leq \log^2(n')$ for every $i \in [n^d]$, we will, with probability $1 - o_n(1)$, construct a sample $S'$ that is contained in $A$, such that if $S$ is scattered then $S'$ is scattered, and if $S$ is $\mathcal{H}^{n,\log^2(n')}_{\mathrm{sign-cnn}}$-realizable then $S'$ is $\mathcal{D}^{n'+1}_z$-realizable. Note that our reduction is allowed to fail with probability $o_n(1)$. Indeed, distinguishing scattered from realizable requires success with probability $\frac{3}{4} - o_n(1)$ and therefore reductions between such problems are not sensitive to a failure with probability $o_n(1)$.

Let $\mathbf{z} = (z_1, \ldots, z_{n'+1})$ where each $z_i$ is drawn i.i.d. from $\mathcal{D}_z$. Let $M = \mathrm{diag}(\mathbf{z})$ be a diagonal matrix. Note that $M$ is invertible with probability $1 - o_n(1)$, since for every $i \in [n'+1]$ we have $Pr_{z_i \sim \mathcal{D}_z}(z_i = 0) \leq Pr_{z_i \sim \mathcal{D}_z}(|z_i| < f(n')) = o(\frac{1}{n'})$. Now, for every $\mathbf{x}_i$ from $S$, denote $\mathbf{x}_i = (\mathbf{x}^i_1, \ldots, \mathbf{x}^i_{\log^2(n')})$ where for every $j$ we have $\mathbf{x}^i_j \in \mathbb{R}^{n'+1}$. Let $\mathbf{x}'_i = (M^{-1}\mathbf{x}^i_1, \ldots, M^{-1}\mathbf{x}^i_{\log^2(n')})$, and let $S' = \{(\mathbf{x}'_1, y_1), \ldots, (\mathbf{x}'_{n^d}, y_{n^d})\}$. Note that if $S$ is scattered then $S'$ is also scattered. If $S$ is realizable by a CNN $h^n_{\mathbf{w}} \in \mathcal{H}^{n,\log^2(n')}_{\mathrm{sign-cnn}}$, then let $\mathbf{w}' = M\mathbf{w}$. Note that $S'$ is realizable by $h^n_{\mathbf{w}'}$. Indeed, for every $i$ and $j$ we have $\langle \mathbf{w}', M^{-1}\mathbf{x}^i_j \rangle = \mathbf{w}^\top M^\top M^{-1}\mathbf{x}^i_j = \mathbf{w}^\top MM^{-1}\mathbf{x}^i_j = \langle \mathbf{w}, \mathbf{x}^i_j \rangle$. Also, note that since $\mathbf{w} \in \{\pm 1\}^{n'+1}$ and $\mathcal{D}_z$ is symmetric, then $\mathbf{w}'$ has the distribution $\mathcal{D}^{n'+1}_z$, and thus $h^n_{\mathbf{w}'}$ is a $\mathcal{D}^{n'+1}_z$-random CNN.

The probability that $\mathbf{z} \sim \mathcal{D}_z^{n'+1}$ has some component $z_i$ with $|z_i| < f(n')$, is at most $(n'+1) \cdot o(\frac{1}{n'}) = o_n(1)$. Therefore, with probability $1 - o_n(1)$ we have for every $i \in [n^d]$ that

$$
\begin{aligned}
\|\mathbf{x}'_i\|^2 &= \sum_{1 \leq j \leq \log^2(n')} \left\| M^{-1} \mathbf{x}^i_j \right\|^2 \leq \sum_{1 \leq j \leq \log^2(n')} \left( \frac{1}{f(n')} \left\| \mathbf{x}^i_j \right\| \right)^2 = \frac{1}{(f(n'))^2} \sum_{1 \leq j \leq \log^2(n')} \left\| \mathbf{x}^i_j \right\|^2 \\
&= \frac{1}{(f(n'))^2} \|\mathbf{x}_i\|^2 \leq \frac{\log^4(n')}{(f(n'))^2} \ .
\end{aligned}
$$

Thus, $\|\mathbf{x}'_i\| \leq \frac{\log^2(n')}{f(n')}$. $\qquad\qquad\qquad\qquad\qquad\qquad\qquad\qquad\qquad\qquad\qquad\qquad\qquad\qquad\qquad$ $\square$

### A.4.2 Proof of Theorem 3.5

Assume that the covariance matrix $\Sigma$ is of size $(n'+1) \times (n'+1)$, and let $n = (n'+1)\log^2(n')$. Note that a $\mathcal{N}(\mathbf{0}, \Sigma)$-random CNN $h^n_{\mathbf{w}}$ has $\log^2(n') = \mathcal{O}(\log^2(n))$ hidden neurons. Let $\mathcal{D}_{\text{vec}}$ be a distribution over $\mathbb{R}^{n'+1}$ such that each component is drawn i.i.d. from $\mathcal{N}(0,1)$. Let $d$ be a fixed integer. By Lemma A.9 and by choosing $f(n') = \frac{1}{n'\log(n')}$, we have that $\text{SCAT}^A_{n^d}(\mathcal{D}_{\text{vec}}, n)$ is RSAT-hard, where $A$ is the ball of radius $n'\log^3(n') \leq n\log(n)$ in $\mathbb{R}^n$. Note that $\mathcal{D}_{\text{vec}} = \mathcal{N}(\mathbf{0}, I_{n'+1})$ ([46]). By Theorem A.1, we need to show that $\text{SCAT}^{A'}_{n^d}(\mathcal{N}(\mathbf{0}, \Sigma), n)$ is RSAT-hard, where $A'$ is the ball of radius $\lambda_{\min}^{-\frac{1}{2}} n\log(n)$ in $\mathbb{R}^n$. We show a reduction from $\text{SCAT}^A_{n^d}(\mathcal{N}(\mathbf{0}, I_{n'+1}), n)$ to $\text{SCAT}^{A'}_{n^d}(\mathcal{N}(\mathbf{0}, \Sigma), n)$.

Let $S = \{(\mathbf{x}_i, y_i)\}^{n^d}_{i=1} \in (\mathbb{R}^n \times \{0,1\})^{n^d}$ be a sample. For every $\mathbf{x}_i$ from $S$, denote $\mathbf{x}_i = (\mathbf{x}^i_1, \ldots, \mathbf{x}^i_{\log^2(n')})$ where for every $j$ we have $\mathbf{x}^i_j \in \mathbb{R}^{n'+1}$. Let $\Sigma = U\Lambda U^\top$ be the spectral decomposition of $\Sigma$, and let $M = U\Lambda^{\frac{1}{2}}$. Recall that if $\mathbf{w} \sim \mathcal{N}(\mathbf{0}, I_{n'+1})$ then $M\mathbf{w} \sim \mathcal{N}(\mathbf{0}, \Sigma)$ ([46]). Let $\mathbf{x}'_i = ((M^\top)^{-1}\mathbf{x}^i_1, \ldots, (M^\top)^{-1}\mathbf{x}^i_{\log^2(n')})$, and let $S' = \{(\mathbf{x}'_1, y_1), \ldots, (\mathbf{x}'_{n^d}, y_{n^d})\}$. Note that if $S$ is scattered then $S'$ is also scattered. If $S$ is realizable by a $\mathcal{N}(\mathbf{0}, I_{n'+1})$-random CNN $h^n_{\mathbf{w}}$, then let $\mathbf{w}' = M\mathbf{w}$. Note that $S'$ is realizable by $h^n_{\mathbf{w}'}$. Indeed, for every $i$ and $j$ we have $\langle \mathbf{w}', (M^\top)^{-1}\mathbf{x}^i_j \rangle = \mathbf{w}^\top M^\top (M^\top)^{-1}\mathbf{x}^i_j = \langle \mathbf{w}, \mathbf{x}^i_j \rangle$. Since $\mathbf{w}' = M\mathbf{w} \sim \mathcal{N}(\mathbf{0}, \Sigma)$, the sample $S'$ is $\mathcal{N}(\mathbf{0}, \Sigma)$-realizable.

We now bound the norms of $\mathbf{x}'_i$ in $S'$. Note that for every $i \in [n^d]$ we have

$$
\begin{aligned}
\|\mathbf{x}'_i\|^2 &= \sum_{1 \leq j \leq \log^2(n')} \left\| (M^\top)^{-1}\mathbf{x}^i_j \right\|^2 = \sum_{1 \leq j \leq \log^2(n')} \left\| U\Lambda^{-\frac{1}{2}}\mathbf{x}^i_j \right\|^2 = \sum_{1 \leq j \leq \log^2(n')} \left\| \Lambda^{-\frac{1}{2}}\mathbf{x}^i_j \right\|^2 \\
&\leq \sum_{1 \leq j \leq \log^2(n')} \left\| \lambda_{\min}^{-\frac{1}{2}}\mathbf{x}^i_j \right\|^2 = \lambda_{\min}^{-1} \sum_{1 \leq j \leq \log^2(n')} \left\| \mathbf{x}^i_j \right\|^2 = \lambda_{\min}^{-1} \|\mathbf{x}_i\|^2 \ .
\end{aligned}
$$

Hence, $\|\mathbf{x}'_i\| \leq \lambda_{\min}^{-\frac{1}{2}} \|\mathbf{x}_i\| \leq \lambda_{\min}^{-\frac{1}{2}} n\log(n)$.

### A.4.3 Proof of Theorem 3.6

Let $n = (n'+1)\log^2(n')$. Let $\mathcal{D}_{\text{vec}}$ be the uniform distribution on $r \cdot \mathbb{S}^{n'}$. Note that a $\mathcal{D}_{\text{vec}}$-random CNN $h^n_{\mathbf{w}}$ has $\log^2(n') = \mathcal{O}(\log^2(n))$ hidden neurons. Let $d$ be a fixed integer. By Theorem A.1, we need to show that $\text{SCAT}^A_{n^d}(\mathcal{D}_{\text{vec}}, n)$ is RSAT-hard, where $A$ is the ball of radius $\frac{\sqrt{n}\log(n)}{r}$ in $\mathbb{R}^n$. By Lemma A.2, the problem $\text{SCAT}^{A'}_{n^d}(\mathcal{H}^{n,\log^2(n')}_{\text{sign}-\text{cnn}})$ where $A'$ is the ball of radius $\log^2(n')$ in $\mathbb{R}^n$, is RSAT-hard. We reduce this problem to $\text{SCAT}^A_{n^d}(\mathcal{D}_{\text{vec}}, n)$. Given a sample $S = \{(\mathbf{x}_i, y_i)\}^{n^d}_{i=1} \in (\mathbb{R}^n \times \{0,1\})^{n^d}$ with $\|\mathbf{x}_i\| \leq \log^2(n')$ for every $i \in [n^d]$, we construct a sample $S'$ that is contained in $A$, such that if $S$ is scattered then $S'$ is scattered, and if $S$ is $\mathcal{H}^{n,\log^2(n')}_{\text{sign}-\text{cnn}}$-realizable then $S'$ is $\mathcal{D}_{\text{vec}}$-realizable.

Let $M$ be a random orthogonal matrix of size $(n'+1) \times (n'+1)$. For every $i \in [n^d]$ denote $\mathbf{x}_i = (\mathbf{x}^i_1, \ldots, \mathbf{x}^i_{\log^2(n')})$ where for every $j$ we have $\mathbf{x}^i_j \in \mathbb{R}^{n'+1}$. For every $i \in [n^d]$ let $\mathbf{x}'_i = $

$(\frac{\sqrt{n'+1}}{r}M\mathbf{x}_1^i,\ldots,\frac{\sqrt{n'+1}}{r}M\mathbf{x}_{\log^2(n')}^i)$, and let $S' = \{(\mathbf{x}_1', y_1),\ldots,(\mathbf{x}_{n^d}', y_{n^d})\}$. Note that if $S$ is scattered then $S'$ is also scattered. If $S$ is realizable by a CNN $h_{\mathbf{w}}^n \in \mathcal{H}_{\text{sign}-\text{cnn}}^{n,\log^2(n')}$, then let $\mathbf{w}' = \frac{r}{\sqrt{n'+1}}M\mathbf{w}$. Note that $S'$ is realizable by $h_{\mathbf{w}'}^n$. Indeed, for every $i$ and $j$ we have

$$\langle \mathbf{w}', \frac{\sqrt{n'+1}}{r}M\mathbf{x}_j^i\rangle = \mathbf{w}^\top \frac{r}{\sqrt{n'+1}}M^\top \frac{\sqrt{n'+1}}{r}M\mathbf{x}_j^i = \langle \mathbf{w}, \mathbf{x}_j^i\rangle\ .$$

Also, note that since $\|\mathbf{w}\| = \sqrt{n'+1}$ and $M$ is orthogonal, $\mathbf{w}'$ is a random vector on the sphere of radius $r$ in $\mathbb{R}^{n'+1}$, and thus $h_{\mathbf{w}'}^n$ is a $\mathcal{D}_{\text{vec}}$-random CNN.

Since $M$ is orthogonal then for every $i \in [n^d]$ we have

$$
\begin{aligned}
\|\mathbf{x}_i'\|^2 &= \sum_{1\le j\le \log^2(n')}\left\|\frac{\sqrt{n'+1}}{r}M\mathbf{x}_j^i\right\|^2 = \frac{n'+1}{r^2}\sum_{1\le j\le \log^2(n')}\|\mathbf{x}_j^i\|^2 \\
&= \frac{n'+1}{r^2}\cdot\|\mathbf{x}_i\|^2 \le \frac{(n'+1)\log^4(n')}{r^2} \le \frac{n\log^2(n)}{r^2}\ .
\end{aligned}
$$

Hence $\|\mathbf{x}_i'\| \le \frac{\sqrt{n}\log(n)}{r}$.

## B   From $\text{CSP}_{n^d}^{\text{rand}}(\text{SAT}_K)$ to $\text{CSP}_{n^{d-1}}^{\text{rand}}(T_{K,q(n)}, \neg T_{K,q(n)})$ ([17])

We outline the main ideas of the reduction.

First, we reduce $\text{CSP}_{n^d}^{\text{rand}}(\text{SAT}_K)$ to $\text{CSP}_{n^{d-1}}^{\text{rand}}(T_{K,q(n)})$. This is done as follows. Given an instance $J = \{C_1,\ldots,C_{n^d}\}$ to $\text{CSP}(\text{SAT}_K)$, by a simple greedy procedure, we try to find $n^{d-1}$ disjoint subsets $J_1',\ldots,J_{n^{d-1}}' \subset J$, such that for every $t$, the subset $J_t'$ consists of $q(n)$ constraints and each variable appears in at most one of the constraints in $J_t'$. Now, from every $J_t'$ we construct a $T_{K,q(n)}$-constraint that is the conjunction of all constraints in $J_t'$. If $J$ is random, this procedure will succeed w.h.p. and will produce a random $T_{K,q(n)}$-formula. If $J$ is satisfiable, this procedure will either fail or produce a satisfiable $T_{K,q(n)}$-formula.

Now, we reduce $\text{CSP}_{n^{d-1}}^{\text{rand}}(T_{K,q(n)})$ to $\text{CSP}_{n^{d-1}}^{\text{rand}}(T_{K,q(n)}, \neg T_{K,q(n)})$. This is done by replacing each constraint, with probability $\frac{1}{2}$, with a random $\neg P$ constraint. Clearly, if the original instance is a random instance of $\text{CSP}_{n^{d-1}}^{\text{rand}}(T_{K,q(n)})$, then the produced instance is a random instance of $\text{CSP}_{n^{d-1}}^{\text{rand}}(T_{K,q(n)}, \neg T_{K,q(n)})$. Furthermore, if the original instance is satisfied by the assignment $\psi \in \{\pm 1\}^n$, the same $\psi$, w.h.p., will satisfy all the new constraints. The reason is that the probability that a random $\neg T_{K,q(n)}$-constraint is satisfied by $\psi$ is $1 - (1 - 2^{-K})^{q(n)}$, and hence, the probability that all new constraints are satisfied by $\psi$ is at least $1 - n^{d-1}(1 - 2^{-K})^{q(n)}$. Now, since $q(n) = \omega(\log(n))$, the last probability is $1 - o_n(1)$.

For the full proof see [17].

## C   Improving the bounds on the support of $\mathcal{D}$ in the convolutional networks

We show that by increasing the number of hidden neurons from $\mathcal{O}(\log^2(n))$ to $\mathcal{O}(n)$ we can improve the bounds on the support of $\mathcal{D}$. Note that our results so far on learning random CNNs, are for CNNs with input dimension $n = \mathcal{O}(t\log^2(t))$ where $t$ is the size of the patches. We now consider CNNs with input dimension $\tilde{n} = t^c$ for some integer $c > 1$. Note that such CNNs have $t^{c-1} = \mathcal{O}(\tilde{n})$ hidden neurons.

Assume that there is an efficient algorithms $\mathcal{L}'$ for learning $\mathcal{D}_{\text{vec}}$-random CNNs with input dimension $\tilde{n} = t^c$, where $\mathcal{D}_{\text{vec}}$ is a distribution over $\mathbb{R}^t$. Assume that $\mathcal{L}'$ uses samples with at most $\tilde{n}^d = t^{cd}$ inputs. We show an algorithm $\mathcal{L}$ for learning a $\mathcal{D}_{\text{vec}}$-random CNN $h_{\mathbf{w}}^n$ with $n = \mathcal{O}(t\log^2(t))$. Let $S = \{(\mathbf{x}_1, h_{\mathbf{w}}^n(\mathbf{x}_1)),\ldots,(\mathbf{x}_{n^{cd}}, h_{\mathbf{w}}^n(\mathbf{x}_{n^{cd}}))\}$ be a sample, and let $S' = \{(\mathbf{x}_1', h_{\mathbf{w}}^n(\mathbf{x}_1)),\ldots,(\mathbf{x}_{n^{cd}}', h_{\mathbf{w}}^n(\mathbf{x}_{n^{cd}}))\}$ where for every vector $\mathbf{x} \in \mathbb{R}^n$, the vector $\mathbf{x}' \in \mathbb{R}^{\tilde{n}}$ is obtained from $\mathbf{x}$ by padding it with zeros. Thus, $\mathbf{x}' = (\mathbf{x}, 0,\ldots,0)$. Note that $n^{cd} > \tilde{n}^d$. Also, note

that for every $i$ we have $h_{\mathbf{w}}^n(\mathbf{x}_i) = h_{\mathbf{w}}^{\tilde{n}}(\mathbf{x}_i')$. Hence, $S'$ is realizable by the CNN $h_{\mathbf{w}}^{\tilde{n}}$. Now, given $S$, the algorithm $\mathcal{L}$ runs $\mathcal{L}'$ on $S'$ and returns an hypothesis $h(\mathbf{x}) = \mathcal{L}'(S')(\mathbf{x}')$.

Therefore, if learning $\mathcal{D}_{\mathrm{vec}}$-random CNNs with input dimension $n = \mathcal{O}(t \log^2(t))$ is hard already if the distribution $\mathcal{D}$ is over vectors of norm at most $g(n)$, then learning $\mathcal{D}_{\mathrm{vec}}$-random CNNs with input dimension $\tilde{n} = t^c$ is hard already if the distribution $\mathcal{D}$ is over vectors of norm at most $g(n) < g(t^2) = g(\tilde{n}^{\frac{2}{c}})$. Hence we have the following corollaries.

**Corollary C.1.** *Let $\mathcal{D}_{\mathrm{vec}}$ be a distribution over $\mathbb{R}^t$ such that each component is drawn i.i.d. from a distribution $\mathcal{D}_z$ over $\mathbb{R}$. Let $n = t^c$ for some integer $c > 1$, and let $\epsilon = \frac{3}{c}$.*

1. *If $\mathcal{D}_z = \mathcal{U}([-r, r])$, then learning a $\mathcal{D}_{\mathrm{vec}}$-random CNN $h_{\mathbf{w}}^n$ (with $\mathcal{O}(n)$ hidden neurons) is RSAT-hard, already if $\mathcal{D}$ is over vectors of norm at most $\frac{n^\epsilon}{r}$.*

2. *If $\mathcal{D}_z = \mathcal{N}(0, \sigma^2)$, then learning a $\mathcal{D}_{\mathrm{vec}}$-random CNN $h_{\mathbf{w}}^n$ (with $\mathcal{O}(n)$ hidden neurons) is RSAT-hard, already if $\mathcal{D}$ is over vectors of norm at most $\frac{n^\epsilon}{\sigma}$.*

**Corollary C.2.** *Let $\Sigma$ be a positive definite matrix of size $t \times t$, and let $\lambda_{\min}$ be its minimal eigenvalue. Let $n = t^c$ for some integer $c > 1$, and let $\epsilon = \frac{3}{c}$. Then, learning a $\mathcal{N}(\mathbf{0}, \Sigma)$-random CNN $h_{\mathbf{w}}^n$ (with $\mathcal{O}(n)$ hidden neurons) is RSAT-hard, already if the distribution $\mathcal{D}$ is over vectors of norm at most $\frac{n^\epsilon}{\sqrt{\lambda_{\min}}}$.*

**Corollary C.3.** *Let $\mathcal{D}_{\mathrm{vec}}$ be the uniform distribution over the sphere of radius $r$ in $\mathbb{R}^t$. Let $n = t^c$ for some integer $c > 1$, and let $\epsilon = \frac{2}{c}$. Then, learning a $\mathcal{D}_{\mathrm{vec}}$-random CNN $h_{\mathbf{w}}^n$ (with $\mathcal{O}(n)$ hidden neurons) is RSAT-hard, already if the distribution $\mathcal{D}$ is over vectors of norm at most $\frac{n^\epsilon}{r}$.*

As an example, consider a CNN $h_{\mathbf{w}}^n$ with $n = t^c$. Note that since the patch size is $t$, then each hidden neuron has $t$ input neurons feeding into it. Consider a distribution $\mathcal{D}_{\mathrm{vec}}$ over $\mathbb{R}^t$ such that each component is drawn i.i.d. by a normal distribution with $\sigma = \frac{1}{\sqrt{t}}$. This distribution corresponds to the standard Xavier initialization. Then, by Corollary C.1, learning a $\mathcal{D}_{\mathrm{vec}}$-random CNN $h_{\mathbf{w}}^n$ is RSAT-hard, already if $\mathcal{D}$ is over vectors of norm at most $n^{\frac{3}{c}} \sqrt{t} = n^{\frac{3}{c}} \cdot n^{\frac{1}{2c}}$. By choosing an appropriate $c$, we have that learning a $\mathcal{D}_{\mathrm{vec}}$-random CNN $h_{\mathbf{w}}^n$ is RSAT-hard, already if $\mathcal{D}$ is over vectors of norm at most $\sqrt{n}$.

Finally, note that Corollary 3.4 holds also for the values of $n$ and the bounds on the support of $\mathcal{D}$ from Corollaries C.1, C.2 and C.3.

## Footnotes

[3] As in $\mathrm{CSP}_{m(n)}^{\mathrm{rand}}(P)$, in order to succeed, and algorithm must return "satisfiable" w.p. at least $\frac{3}{4} - o_n(1)$ on every satisfiable formula and "random" w.p. at least $\frac{3}{4} - o_n(1)$ on random formulas.