[Reviews · NeurIPS 2020]

Review 1

Summary and Contributions: In this paper the authors consider the following question: what is an important aspect of a neural network that makes it work well in practice when it's known to be theoretically hard to learn neural networks. There have been many prior works in this direction but most works have proved hardness of learning neural networks under strange architectures or distributions. In this paper their main contribution is to look at "natural" distributions which include for example the uniform distribution and normal distribution and show hardness of learning neural networks even under such natural distributions.

Strengths: Given how important neural networks are in practice, understanding their theoretical underpinning is an important question in ML and this paper shows yet another direction in which it is hard to learn NNs. This will definitely be of interest to the learning theory community in NeurIPS. The main selling point is that prior works looked at arbitrary weights, solely with the purpose of proving lower bounds but in this paper the authors look at natural weights (under which one might have expected to prove positive results) and show that even in this setting, NNs are hard to learn. As is expected in hardness of learning results, the result of this paper conditional and the authors show the hardness of learning assuming that refuting a random K-SAT instance is hard (which is widely believed to be true).

Weaknesses: To be honest, it is not entirely surprising to me that there is a hardness result for learning depth-2 NNs under the uniform distribution assuming RSAT. My intuition comes from the fact that we believe even TC_0^2 (i.e., depth 2 threshold circuits which are supposed to model NNs) are believed to be hard to learn under the uniform distribution. So in that aspect, i believe this result isn't too surprising. As I mention later, another weakness is, it is not at all clear to me how the techniques in this paper differ from many of the prior works that also establish the hardness of NNs under the RSAT distribution (I agree the result is stronger and nicer here, but about the techniques, I'm not sure).

Correctness: I verified only a few claims. Given my weakness criticism above that the authors do not make any effort to help navigate this paper from the theorems/lemmas in the 8-page abstract to the supplementary material, it is very hard for me to verify everything.

Clarity: the writing of the paper can be improved significantly (it is written right now as if intentionally wanting to make it look complicated). I can imagine because of the 8-page format it's terse, but if you see page 5 and 6 for example, its just filled with a bunch of theorems and corollaries (with absolutely no reference to where I should find their proofs or even a proof sketch or any indication if this is known/hard to prove and so on). I'd strongly encourage the authors to improve the presentation.

Relation to Prior Work: Not exactly. They state prior works clearly and what is their improvement in terms of results, but in terms of techniques it's not clear to me.

Reproducibility: Yes

Additional Feedback: In this paper the authors consider the following question: what is an important aspect of a neural network that makes it work well in practice when it's known to be theoretically hard to learn neural networks. There have been many prior works in this direction but most works have proved hardness of learning neural networks under strange architectures or distributions. In this paper their main contribution is to look at "natural" distributions which include for example the uniform distribution and normal distribution and show hardness of learning neural networks even under such natural distributions. Given how important neural networks are in practice, understanding their theoretical underpinning is an important question in ML and this paper shows yet another direction in which it is hard to learn NNs. The main selling point is that prior works looked at arbitrary weights, solely with the purpose of proving lower bounds but in this paper the authors look at natural weights (under which one might have expected to prove positive results) and show that even in this setting, NNs are hard to learn. As is expected in hardness of learning results, the result of this paper conditional and the authors show the hardness of learning assuming that refuting a random K-SAT instance is hard (which is widely believed to be true). Technically the main ideas are as follows: In order to model a depth-2 NN, let W be a n x m matrix with m neurons  and suppose this defines the function h_W(x)= [ sum_i [ <w_i, x> ] _+|_[0,1] where the symbols correspond to the ReLU function. These even correspond to convolutional neural networks. Let D_mat be a distribution over such W matrices, supose a learner is given (x, h_W(x)) for x drawn from some distribution D, the learner learns a function  h that approximates h_W well enough under D. In this paper they show hardness of even learning this class h_W under the average case, i.e., for a uniformly random concept in the concept class (in contrast to PAC learning where the complexity is measured according the hardest concept).  In order to achieve hardness of learning random neural networks for a fixed distribution D_mat, the authors find a subclass of {h_W} for a specially designed W such that learning this subclass reduces to randomly learning under the distribution in D_mat. The choice of W is also such that it allows them to construct a set of {h_W'} whose hardness can be reduced to random SAT (this last statement requires some extra work to be done) Overall, here is my impression of this work:Pros: Understanding theoretical hardness of NNs is an important task and in this paper they make an advance towards understanding it under "natural" distributions whereas prior works looked at adverserial distributions with NNs having weights that were huge. Cons: One aspect which seems slightly hidden to me is how does this compare to prior works. I took a look at a few prior works which also exhibit RSAT hardness of learning NNs and the techniques seem similar. I would encourage the authors to clearly state how the technques used here are different than that was used before. To be honest, I find the construction of the special set of W' matrices above, very similar to what is done before when reducing hardness of NNs to hardness of RSAT (could the authors also clarify this). Secondly, the writing of the paper can be improved significantly (it is written right now as if intentionally wanting to make it look complicated). I can imagine because of the 8-page format it's terse, but if you see page 5 and 6 for example, its just filled with a bunch of theorems and corollaries (with absolutely no reference to where I should find their proofs or even a proof sketch or any indication if this is known/hard to prove and so on). I'd strongly encourage the authors to improve the presentation. Post rebuttal: I thank the authors for their response and I upgrade my score from 6 to 7. I'd recommend implementing a certain comments in your final submission for NeurIPS proceedings (if accepted).


Review 2

Summary and Contributions: The paper shows that learning two-layer neural networks with an extremely simple structure: O(\log^2 n) hidden nodes, all-one weights in the top layer, and uniformly random weights for the lower layer, is hard, under the so-called Random-K-SAT hypothesis. The RK-SAT hypothesis states that "random instances" of k-SAT do not admit polynomial time algorithms (in a certain technical sense). This is an interesting result, as it rules out the hypothesis that learning NNs is hard only for "worst case" NNs. That said, it is important to note that the paper shows this hardness under a worst-case _input distribution_ (that depends on the random network). Thus the result can be interpreted as saying that hardness of learning NNs comes more from the worst-case nature of the input than from the worst-case nature of the network weights. The proof is quite nice. It is based on a non-trivial extension of earlier results by Daniely and Shalev-Shwartz. It proceeds by first giving a framework that lets one convert a worst-case hardness result to one for matrices drawn from the desired distribution. This turns out to be possible if the worst-case hardness holds under an appropriately strong structural assumption, that they call sign-CNNs. The hardness of learning this class is proved under the RK-SAT hypothesis, using ideas similar to prior work.

Strengths: The result is nice, and it is of a broad interest to the ML community. Reductions between average problems tend to skew distributions, so it is nice that the paper obtains hardness results for very simple average case models.

Weaknesses: Perhaps the authors should make the point about the worst-case nature of the input distributions more clear. It is by now pretty well-known that structure in inputs is key to modern deep learning, so the paper can be viewed as reinforcing this message.

Correctness: Yes, to the best of my knowledge.

Clarity: Yes, the paper is quite well-written.

Relation to Prior Work: Yes

Reproducibility: Yes

Additional Feedback: Please address the comment on input distribution.


Review 3

Summary and Contributions: This paper proves a series of hardness results for learning two-layer neural networks. Specifically, it shows that if the parameters of the first layer are randomly sampled from some natural distribution (e.g. uniform sphere, Gaussian), and if an adversary can choose a worst-case input distribution, then (improperly) learning the network is RSAT-hard. This result extends the existing hardness result where the network parameters are assumed to be worst-case. Here, the natural distribution assumption is more likely to hold in practice than the worst-case assumption.

Strengths: To the best of my knowledge, this is a novel result on the learnability of neural networks. The idea of constructing the random parameter matrix in Section 4 is interesting. The idea of reducing to the RSAT-hardness may shed light on proving hardness for other kinds of networks.

Weaknesses: 1. The assumption that the adversary can choose a worst-case input distribution is too strong and impractical. Although the paper has used a norm constraint to weaken the adversary, it still feels difficult to connect the hardness result to real applications. 2. The weights on the second layer are all 1. This assumption breaks the "naturalness" of the network. Is this essential or can it be generalized? 3. The result only holds if the hidden dimension m = O(log^2(n)), which makes the result less interesting. Is there a way to prove similar results with more hidden nodes?

Correctness: To the best of my knowledge, the theoretical results are correct.

Clarity: The paper is not hard to follow, but there are way too many concepts and definitions. The authors are encouraged to provide more intuition for each concept.

Relation to Prior Work: Yes

Reproducibility: Yes

Additional Feedback: Overall, this is an interesting theoretical paper, but the I feel that the assumptions (especially the assumption on the input distribution) are way too strong to make it interesting enough for understanding the hardness of learning practical neural networks. After rebuttal: I read the rebuttal. The authors addressed my 2nd and 3rd concern, which makes the paper stronger. Given the satisfying feedback, I'd raise the score from 5 to 6.


Review 4

Summary and Contributions: This paper studies the problem of proving hardness of learning neural networks, in settings where the edge weights follow a natural distribution such as sub-gaussian, uniformly random unit vectors etc. Formally, their setting is the following. Let \cD be a distribution over real valued matrices M = (m_1,...,m_n) where the columns m_i are distributed i.i.d from some natural distribution \cD_{vec}. The matrix M can be used to define a depth-2 neural network h_M(x) = clip_[0,1](\sum_{i \in [n]} <m_i,x>). The main contribution of the paper is the following. With high probability over choices of M \sim \cD, there are no polynomial time algorithms which can learn a predictor h satisfying \Ex_{x \sim \cD}[(h_M(x)- h(x))^2] \leq 1/10, assuming there are no efficient algorithm for refuting random K-SAT formulas. This assumption has been useful in the past for ruling out quanlitatively similar results for learning halfspaces, dnfs etc The reductions in the paper use the following two step approach: 1. Reduce the Random K-SAT refutation problem to that of learning \cH_{DNF}-neural networks, where the columns of M \in \ch_{sign-cnn} are supported on {0,\pm 1} entries. The key observation here is that the inverted TRIBES function (AND of OR's) can be modeled by a depth-2 NN. 2. Reduce learning \cD_{sign-cnn}-neural networks to the problem of learning \cD_{mat}-neural networks, where \cD_{mat} is the target distribution. The first step follows almost immediately from the aforementioned result on random k-sat based hardness of learning dnfs, the second step uses the fact h_M(x) = y iff h_{MW}(W^{-1}x) =y, for any invertible map W. This reduces learning M-neural nets with M \in \cH_{sign-cnn} to the problem of learning MW-distributed neural networks, when W is chosen from an appropriate distribution e.g., if the targed distribution \cD_{mat} requires i.i.d gaussian entries, then sampling W from the gaussian distribution suffices. Some additional work is needed to show invertibility of W and give a high probability bound on the length of vectors output by the reduction; again this follows from tail bounds on singular values.

Strengths: 1. Shows that weight distributions alone do not induce an intractibility barrier for learning depth 2 neural networks, and therefore instances which are easy in practice should satisfy other conditions. This is good progress towards understanding the complexity of learning neural nets. 2. Rules out even improper learning using arbitrary efficiently evaluatable hypothesis classes.

Weaknesses: The results mostly build on the previously known K-SAT based hardness of learning DNFs, (which almost immediately gives hardness of learning structured 2-NNs).

Correctness: The proofs seems to be correct

Clarity: The paper is quite well written, the proof overview section does a good job of giving an informal yet more or less complete description of their proof technique.

Relation to Prior Work: Yes, this is clearly done.

Reproducibility: Yes

Additional Feedback: Some general comments. 1. It would be good to explicitly point out that if Assumption 2.1 is true for all polynomial time algorithms running in time n^C, then your results rule out learning in time n^{f(C)}. 2. Page 6., the notation GL(n) for the matrix group is never introduced, and perhaps can be avoided altogether since it is used only once. 3. Page 7 last para. The definition of S' should have (x'_i, h_W(x'_i)).

[Author Response · NeurIPS 2020]

We thank the reviewers for their efforts. Below we address the main comments.

**Reviewer #1**

The reviewer asked how the techniques in this paper differ from the prior work that established the hardness of neural networks under the RSAT assumption. First, we show that learning neural networks is hard already for networks with a special structure that we call sign-CNN. It requires a reduction from the RSAT problem that is different from the reduction used in prior work. We do use in this reduction a Lemma from [17], so we do not repeat the work that has been done there. Second, we show that the properties of sign-CNNs allow us to transform an unknown sign-CNN to a random network, by a multiplication with a random matrix that has a special structure. This method is new, and does not appear in prior work. Finally, in order to bound the support of the input distribution, we need to analyze the singular values of our special-structure random matrices.

The reviewer also commented on the presentation. We did make much effort to present the results and proofs in a simple and clear way. In the "proof ideas" and "proof structure" sections we give a high-level overview of the proofs. In the appendix, we first establish some needed lemmas, and then each theorem is proved in a separate subsection. Since all theorems require common lemmas and constructions, skipping directly from the "results" section to the subsection in the appendix where the theorem is proved is not possible. Also, for the same reason, giving a sketch of each proof in the "results" section is not possible. Instead, the "proof ideas" section is essentially a sketch of the proofs.

**Reviewer #2**

The reviewer asked about the worst-case nature of the input distribution. Prior hardness results for learning neural networks assume that both the input distribution and the weights are worst-case. We show that the problem is hard already for simple networks with natural weights, but the worst-case nature of the input distribution remains. Thus, we show that even very strong assumptions on the network are not sufficient for efficient learning, and therefore that assumptions on the input distribution are necessary. We believe that positive results on the learnability of neural networks would require a combination of assumptions on the weights and on the input distribution.

**Reviewer #4**

The reviewer raised three concerns:

1. "The assumption that the adversary can choose a worst-case input distribution is too strong and impractical":
   The standard PAC-learning framework requires an algorithm that learns successfully for every input distribution and every hypothesis in the class. Hence, the assumption that the adversary can choose a worst-case input distribution is the standard assumption in PAC learning. While prior hardness results for learning neural networks assume worst-case input distribution and worst-case weights, we show that the problem is hard already for networks with natural weights (but the worst-case nature of the input distribution remains). The practical importance of our result, is that it suggests that in order to establish efficient algorithms for learning neural networks, or to show polynomial-time guarantees for existing algorithms, assumptions on the network's architecture and weights are not sufficient, and assumptions on the input distribution are necessary. Hence, our result may help focussing the efforts on directions where positive results are possible. We believe that a combination of assumptions on the weights and on the input distribution is necessary in order to obtain positive results.

2. "The weights on the second layer are all $1$. This assumption breaks the "naturalness" of the network. Is this essential or can it be generalized?":
   The choice to focus on networks where the weights in the second layer are all $1$ is not essential. Our results hold also where the weights in the second layer have other fixed values, and also in the case where they are random. We may add a remark on this point.

3. "The result only holds if the hidden dimension $m = O(log^2(n))$, which makes the result less interesting. Is there a way to prove similar results with more hidden nodes?":
   The results do extend to more hidden nodes. As we mention in page 4 (lines 168-170), the results hold (with minor changes) for any $m = \omega(log(n))$, including networks with many hidden neurons. We chose to focus on $m = O(log^2(n))$ since we wanted to show hardness of learning already where the number of hidden neurons is relatively small.

**Reviewer #5**

We thank the reviewer for his/her comments and will fix accordingly.

[Meta-Review · NeurIPS 2020]

The paper received four reviews. Initially, the scores were borderline but more on the side of accepting. The reviews point out that the paper provides a new style of results on a fundamentally important problem, with interesting techniques. However, one reviewer commented on the result itself being unsurprising; two reviewers were critical of the fact that the result depends on a worst-case input distribution; and some concern was expressed about relation to previous work. The reply from the authors was considered by the reviewers mainly satisfactory, but only partially so regarding the worst-case input distribution. After discussion, some scores were raised and the reviewers are in agreement about recommending accepting the paper.